# EAT: Entropy After </Think> for Reasoning Model Early Exiting

## Abstract

Large reasoning models show improved performance with longer chains of thought. However, recent work has highlighted (qualitatively) their tendency to overthink, continuing to revise answers even after reaching the correct solution. We quantitatively confirm this inefficiency by tracking Pass@1 for answers averaged over a large number of rollouts and find that the model often begins to always produce the correct answer early in the reasoning, making extra reasoning a waste of tokens. To detect and prevent overthinking, we propose a simple and inexpensive novel signal—Entropy After </Think> (EAT)—for monitoring and deciding whether to exit reasoning early. By appending a stop thinking token (</think>) and monitoring the entropy of the following token as the model reasons, we obtain a trajectory that decreases and stabilizes when Pass@1 plateaus; thresholding its variance under an exponential moving average yields a practical stopping rule. Importantly, our approach enables adaptively allocating compute based on the EAT trajectory, allowing us to spend compute in a more efficient way compared with fixing the token budget for all questions. Empirically, on MATH-500 and AIME-2025, EAT reduces token usage by 13–21% without harming accuracy, and it remains effective in black-box settings where logits from the reasoning model are not accessible, and EAT is computed with proxy models.

## 1 Introduction

Large reasoning models such as GPT o1 (Jaech et al., 2024), DeepSeek R1 (DeepSeek-AI et al., 2025), or S1 (Muennighoff et al., 2025) exhibit the test-time scaling phenomenon (Snell et al., 2024; Wu et al., 2024): when given more computation budget at inference, they generate longer chains of thought and achieve higher accuracy on reasoning tasks. This suggests that models can effectively improve performance by "thinking longer," even without additional training.

However, current practice allocates a fixed token budget per question, regardless of difficulty. Easy questions can be solved confidently with just a few or no reasoning steps, while harder ones require extended deliberation. This uniform allocation wastes compute on simple instances while potentially underserving complex ones. This manifests as overthinking (Chen et al., 2024; Sui et al., 2025), where models unnecessarily revise correct reasoning, as illustrated in Appendix H. Quantitatively, we observe that Pass@1 accuracy (averaged over multiple random rollouts, Eq. (8)) often stabilizes and saturates early in the reasoning chain, sometimes even within the *first 10-20%* of the allocated token budget (Fig. 1, top row). This indicates that the model continues to generate the same answer regardless of additional reasoning steps, making further computation redundant.

This observation highlights the need for adaptive test-time computation mechanisms, where the model dynamically determines how much reasoning to allocate to each input. A natural principle for enabling such adaptivity is uncertainty: under submodularity assumptions, computation should continue as long as uncertainty decreases and stop once the uncertainty has stabilized. This idea has a long history in adaptive computing, from dynamically determining the number of steps in RNN (Graves, 2016) to confidence-based early exiting rules in deep networks (Teerapittayanon et al., 2016; Schuster et al., 2022). In the context of reasoning models, however, it is not clear whether there exist effective and inexpensive uncertainty signals for this purpose.

In this work, we propose Entropy After </Think> (EAT) as such a signal (Sec. 4.1). By appending a stop thinking token (</think>) after each generated reasoning line and measuring the entropy of the model's next-token distribution over the single token after </think>, we obtain a quantity

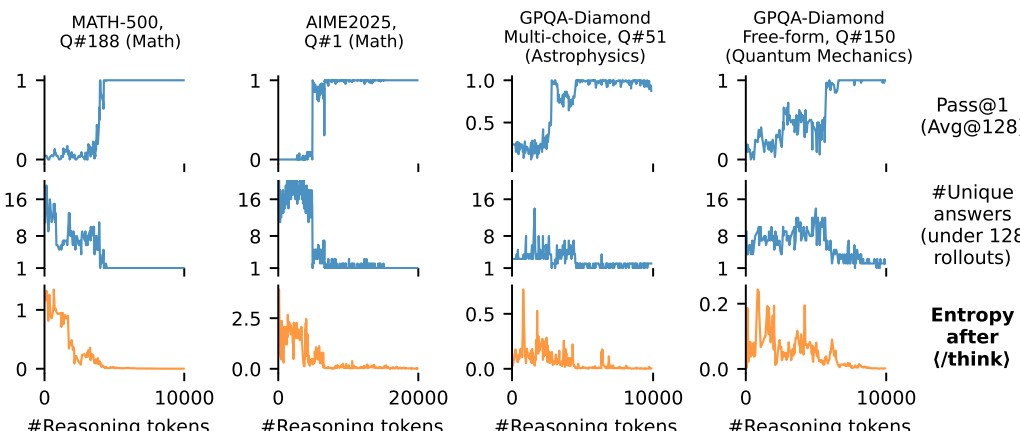

Figure 1: **EAT provides an informative signal to prevent overthinking in reasoning models.** We evaluate questions from four datasets (columns) using DeepSeek-R1-0528-Qwen3-8B, where we plot different metrics against the number of reasoning tokens. The first row shows that Pass@1 averaged over 128 rollouts (Eq. (8)) quickly saturates, indicating overthinking in reasoning. The number of unique answers under multiple rollouts (second row) stabilizes near one when Pass@1 converges; however, its evaluation has a high and non-deterministic overhead. We propose to manually append the stop thinking token (</think>) during reasoning and look at the **E**ntropy over the single token **A**fter </**T**hink> (EAT, bottom row, Eq. (5)), which drops and stabilizes at the point where Pass@1 plateaus, providing a cheap and deterministic signal for early exiting.

that sharply decreases and stabilizes at precisely the point where accuracy saturates (Fig. 1, bottom row). When monitored over time with an exponential moving average, the variance of EAT provides a reliable criterion for early exiting (Alg. 1, Sec. 4.2).

In summary, this work makes three key contributions. (i) We provide the first quantitative demonstration that reasoning models overthink from a distribution dynamics perspective, motivating the usage of uncertainty as an early exit criterion. (ii) We introduce Entropy After </Think> (EAT), a lightweight informative signal for early exiting, and propose a practical variance-based stopping rule using an exponential moving average. (iii) We empirically validate the effectiveness of EAT on benchmarks including MATH-500 and AIME-2025, showing that it reduces token usage by up to 21% without sacrificing accuracy (Sec. 5). Crucially, EAT remains effective even when the reasoning model's logits are inaccessible, where we compute EAT using a smaller proxy reasoning model that only observes the reasoning text from the black-box reasoning model, making our method applicable to commercial APIs that do not expose full token probabilities. To support future research, we also release large-scale answer rollouts and intermediate reasoning traces (taking over 20K GPU hours of computation), enabling future works to study early exiting without re-running costly experiments.

## 2 RELATED WORK

**Early exiting of deep learning models** In deep learning literature, it is common to perform layer-wise early exiting to prevent overthinking (Kaya et al., 2019) and improve efficiency. Using confidence / uncertainty as a metric for early exiting is a widely adopted wisdom. In particular, it has been shown that confidence scores can early exit CNNs, diffusion or LLMs (Graves, 2016; Teerapittayanon et al., 2016; Elbayad et al., 2019; Schuster et al., 2022; Jazbec et al., 2024). Our approach shares a similar spirit in that we also propose to use entropy, an uncertainty-related metric, as an early exiting criterion, but for reasoning process.

**Early exiting of long chain of thought** Since the release of open-source reasoning models, in particular, the DeepSeek R1 series, many recent pre-prints study early exiting. These methods share similiar principles: they generate an answer rollout string during reasoning and look at various properties of the rollout related to uncertainty. Yang et al. (2025) look at confidence score, measured by length-normalized log-likelihood (Malinin & Gales, 2020) of all tokens in the answer. Liu & Wang (2025); Fu et al. (2025); Mao et al. (2025) study whether the answer has stopped changing. Yong et al. (2025) look at the entropy empirically estimated through generating multiple rollouts

using beam search. Generating a rollout is an expensive operation and is non-trivial in the sense that it often involves additional decoding hyperparameters and engineering techniques for efficiency. For example, Yang et al. (2025) use branch-parallel decoding and only evaluate the metric when special keywords (such as "wait") pop up for efficiency. Liu & Wang (2025) perform a tree-based search over completions in order to estimate the entropy. In contrast, we demonstrate that the entropy over just one single token after `</think>` is informative and rollout may not be necessary. This results in large efficiency gains. Additionally, Zhang et al. (2025) train a classifier for probing a reasoning model's hidden states to predict when to exit, which differentiates from `EAT`, which does not require any training. Lastly, unlike previous works, we demonstrate compatibility with black-box models: we can perform early exiting via only monitoring the verbal output of the reasoning model with a white-box proxy model without using the black-box reasoning model's internal information.

**Reasoning and entropy** Entropy plays an important role in understanding the reasoning process. Ton et al. (2025) studies LLM reasoning from an information theory perspective; they made arguments similar to our work in that they suggest information gain / uncertainty should gradually increase / decrease throughout the reasoning process. However, the main difference is that they trained a separate LLM to estimate the answer given the reasoning chain and look at the entropy provided by this separate LLM, whereas we directly look at the entropy after `</think>` token using an out-of-the-box reasoning model without any fine-tuning / training. Wang et al. (2025) and Cheng et al. (2025) study token-level entropy inside reasoning chains to improve the post-training, in particular, they propose to apply policy gradient only on high entropy tokens and include entropy regularization in the objective to encourage reasoning process's exploration, respectively.

## 3 BACKGROUND: LARGE LANGUAGE MODELS AND REASONING MODELS

### 3.1 LARGE LANGUAGE MODELS

A decoder-only large language model (LLM) is a next token prediction model. Given a pre-trained LLM denoted as $\theta$, with a vocabulary of $|V|$ tokens, it takes in a context string `C` and produces a predictive distribution over the next token after `C`, denoted as

$$f(\texttt{C};\theta) = \boldsymbol{p} \in \Delta^{|V|-1}, \tag{1}$$

where $\Delta^{|V|-1} = \left\{ \boldsymbol{p} \in \mathbb{R}^{|V|} \;\middle|\; \boldsymbol{p}_i \geq 0 \;\; \forall i, \;\; \sum_{i=1}^{|V|} \boldsymbol{p}_i = 1 \right\}$ denotes a $|V|-1$ dimensional simplex. The entropy of this discrete distribution is,

$$\mathbb{H}(f(\texttt{C};\theta)) = \sum_{i=1}^{|V|} -\boldsymbol{p}_i \log \boldsymbol{p}_i, \tag{2}$$

a strictly positive quantity that measures the uncertainty over the next token.

Now given a context `C`, in order to generate a completion using an LLM, we iteratively sample the next token from $f(\texttt{C};\theta)$ using certain strategies such as greedy decoding, random sampling, etc. after which the newly generated token will be added to `C` and we continue this iterative process until either a token budget is hit or some special tokens are generated. For the purpose of notation convenience, we define two special decoding operations

$$\texttt{GenerateNewLine}(\cdot;\theta), \qquad \texttt{GenerateTillEoS}(\cdot;\theta), \tag{3}$$

which denotes generating new tokens till a paragraph separation character `\n\n` is hit or End of Sequence token `<EoS>` is generated, respectively. By default, we apply random sampling for both operations unless a different decoding strategy is explicitly specified.

### 3.2 REASONING MODELS

Recently, it was shown that LLMs can perform reasoning after dedicated post-training (DeepSeek-AI et al., 2025). Reasoning models often generate very long chains of thought (more than 5K tokens), much longer than those generated by prompting the model to think step by step (Wei et al., 2022), often yielding a few hundred tokens.

Post-training encourages the outputs to be structured in a special format via reward design. To be more specific, it is often structured as

$$\texttt{Q, <think>,}\underbrace{\texttt{r}_1,\ldots,\texttt{r}_n,\ldots,\texttt{r}_N}_{\texttt{R}}\texttt{,</think>,The final answer is:, A} \tag{4}$$

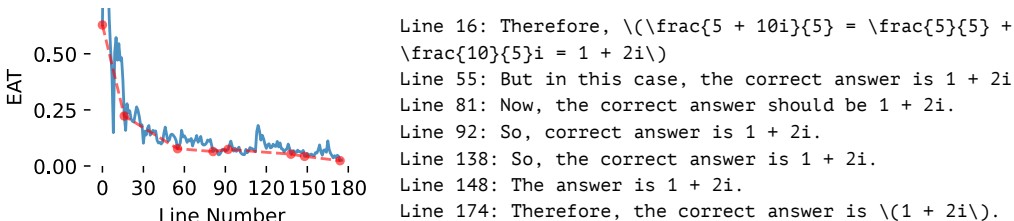

Line 16: Therefore, \(\frac{5 + 10i}{5} = \frac{5}{5} + \frac{10}{5}i = 1 + 2i\)
Line 55: But in this case, the correct answer is 1 + 2i.
Line 81: Now, the correct answer should be 1 + 2i.
Line 92: So, correct answer is 1 + 2i.
Line 138: So, the correct answer is 1 + 2i.
Line 148: The answer is 1 + 2i.
Line 174: Therefore, the correct answer is \(1 + 2i\).

Figure 2: **`EAT` shows a monotonically decreasing pattern every time a conclusion is reached.** Intuitively, since `EAT` is related to information gain (Eq. (6)), we hypothesize that `EAT` will monotonically decrease at each reasoning step. However, in our experiments, since it is hard to know when a step has begun or ended, we evaluate `EAT` every line, and the `EAT` trajectory shows non-smooth patterns with lots of small bumps in the middle (blue line). However, if we only look at the `EAT` values at each line where an answer is drawn (red dots, exact text shown on the right, manually annotated), which we can consider as a "step", `EAT` trajectory shows a smoother decreasing pattern.

where commas denote the concatenation of the strings; Q and A denote the question and the generated answer, and `<think>` and `</think>` are special tokens denoting the beginning and end of thinking introduced during post-training. Between the begin / stop thinking token is the reasoning process R, where each $r_n$ inside R denotes a line (ended with `\n\n`) in the reasoning process. If we read the reasoning lines verbally, several lines can compose a reasoning step, which constructs a complete step of self-revision, error-correction, etc. In practice, reasoning LLM shows the ability of "test time scaling": the more token budget we allocate for R, the higher the chance of getting a correct answer.

### 3.3 REASONING MODELS OVERTHINK

One of the central goals of the post-training is to enable the LLM to generate longer responses to all questions. In fact, the response length is considered an important metric to monitor the progress of post-training, e.g. Fig. 3 in DeepSeek-AI et al. (2025), and in the model card of the latest DeepSeek release[1], the authors contrast it with an older release by how many more tokens it uses per question.

Despite long chains of thought improving performance, for many questions, especially the easy ones, a big portion of the reasoning is redundant. The top row in Fig. 1 illustrates such overthinking behavior. After each reasoning line, we force the model to roll out an answer and measure the portion of correct answers under 128 randomly generated answer strings (Pass@1 Avg@128, Eq. (8)). We find that in the reasoning process, the metrics often stabilize at an early phase in the reasoning chain, long before the token budgets are used up or `</think>` is generated. Practically, this implies that there is no point in continuing to generate reasoning steps, as the model will be capable of getting a correct answer 100 percent of the time, at which point we can just stop thinking and elicit an answer.

Overthinking causes a significant cost as the overhead of LLM outputs per token is much higher than that of the input tokens, due to the un-vectorizable and sequential nature of LLM decoding.[2] It is also indicative of a broader problem, namely the need for more introspective capabilities for reasoning. This motivates us to develop methods for monitoring and early exiting the reasoning process, allowing us to halt the thinking process when the performance gain has saturated.

## 4 METHOD

In this section, we introduce Entropy After `</Think>` (`EAT`) in Sec. 4.1, a quantity to monitor the reasoning process and determine early exiting. We then discuss early exiting with `EAT` in Sec. 4.2 then efficiency considerations in Sec. 4.3

### 4.1 ENTROPY AFTER `</THINK>`

Given a reasoning LLM $\theta$, we study the following quantity,

$$\texttt{EAT} = \mathbb{H}(f(\texttt{Q}, \texttt{<think>}, \texttt{r}_1, \dots, \texttt{r}_n, \texttt{</think>}, \texttt{\n}; \theta)) \quad (5)$$

---

[1] https://huggingface.co/deepseek-ai/DeepSeek-R1-0528

[2] The overhead differences are also reflected by the higher cost-per-token for output as compared to input token in commercial LLM APIs.

where $f(\cdot)$ and $\mathbb{H}(\cdot)$ are the next token prediction probability and entropy defined in Eq. (1) and Eq. (2). The motivation behind EAT is to study the **information gain** from the reasoning process. The information gain of a reasoning process $\texttt{R} = \texttt{r}_1, \ldots, \texttt{r}_n$ is the reduction in uncertainty, measured by the difference in entropy between the next-token distribution before and after reasoning,

$$\text{Information Gain} = \underbrace{\mathbb{H}\left(f\left(\texttt{Q}, \texttt{<think>}, \texttt{</think>}, \texttt{\textbackslash n}; \theta\right)\right)}_{\text{Constant w.r.t reasoning steps}}$$
$$- \underbrace{\mathbb{H}\left(f\left(\texttt{Q}, \texttt{<think>}, \texttt{r}_1, \ldots, \texttt{r}_n, \texttt{</think>}, \texttt{\textbackslash n}; \theta\right)\right)}_{\text{EAT}}. \tag{6}$$

Since the first term does not depend on the exact reasoning steps, EAT describes the information gain from R.

It is worth emphasizing that EAT looks at only the entropy over one token; this differs from prior work that requires generating a full answer after appending `</think>`. Surprisingly, the entropy over just one token is highly informative. To illustrate this by way of example, we visualize the trace of EAT in the bottom row in Fig. 1. The EAT trajectory begins at a high value and gradually decreases and stabilizes, indicating that the single-token information gain gradually increases and then stabilizes. Importantly, when EAT stabilizes, the model's performance, measured by Pass@1 (Eq. (8)), also stabilizes. This correlation inspires our central idea to perform early exiting of the reasoning chain with EAT when it stabilizes. Note that EAT contrasts with the entropy inside the chain of thought, which shows a much noisier pattern (Appendix D) and unclear correlation with Pass@1. Additionally, we observe that for older models, a prefix string needs to be included after `</think>` in order for EAT to be informative; we provide a detailed discussion in Appendix B.

Ideally, we would like to study how EAT changes after every step of reasoning. Therefore, we manually annotate a reasoning trajectory where we mark the position where an answer is provided, and we consider these positions as the end of a reasoning step. We track the values of EAT at these positions and the result is presented in Fig. 2: The trajectory at these positions shows a less noisy and more monotonically decreasing pattern, indicating each reasoning step reduces uncertainty.

## 4.2 AUTOMATIC EARLY EXITING

Fig. 1 shows that when EAT stabilizes, the performance gain from reasoning also saturates, which motivates us to develop an EAT-based criterion for automatic stopping. Alg. 1 shows the detailed end-to-end algorithm we propose. In particular, we look at the *exponential moving average* (EMA) estimation (Bruce, 1969) of EAT trajectory's variance. We maintain a running mean ($\hat{M}$) and variance ($\hat{V}$), and we update them every time a new reasoning line is generated, upon which a new value of EAT is computed:

$$\hat{M} = (1 - \alpha)\hat{M} + \alpha\texttt{EAT}, \qquad \hat{V} = (1 - \alpha)\hat{V} + \alpha(\texttt{EAT} - \hat{M})^2, \tag{7}$$

where $\alpha \in (0, 1)$ is the timescale parameter that controls the "window size" of the moving average. In practice, we find that $\alpha \approx 0.2$ works well for most problems (Fig. 10, Appendix G.2). Intuitively, $\hat{V}$ approximately measures the EAT's variance over the past $1/\alpha$ iterations, and when the $\hat{V}$ goes below a pre-defined threshold $\delta > 0$, we halt the reasoning process.

The most critical parameter in Alg. 1 is the threshold $\delta$, which allows one to *trade off compute for performance while preserving per-sample adaptivity*. Firstly, smaller values of $\delta$, i.e., a stricter stable condition, often take longer reasoning to achieve and vice versa. However, the actual reasoning budget is *adaptive*: For a fixed $\delta$, it takes less reasoning for EAT to stabilize and $\hat{V}$ to reach $\delta$ on easy problems, therefore fewer tokens will be used. On challenging problems, it takes more reasoning efforts, and Alg. 1 allocates more budgets. We provide examples of the early exiting procedure in Fig. 3, where we visualize traces of EAT, $\hat{V}$, and the early exiting point decided by the threshold $\delta$.

Importantly, our approach also applies in *black-box setting*, where only the reasoning model's verbal output is accessible and its next-token probabilities are not. In such cases, we compute and monitor EAT with a proxy model, denoted as $\phi$ in Alg. 1. This helps in practice as large-scale reasoning models, which often have more than 500B parameters, have to be deployed through a highly complicated pipeline spanned across multiple GPU nodes, where adding extra components can be non-trivial. The main reasoning model can remain massive, while the proxy can be a small 1.5B parameter model running on a single consumer GPU, yet still enable early exiting.

---

**Algorithm 1** Early exiting reasoning process based on variance of `EAT`.

---

**Input:** Question Q, max token limit $T$; Variance threshold $\delta > 0$, EMA timescale $\alpha \in (0, 1)$.
**Input:** Reasoning LLM $\theta$; (*Optional*) A proxy LLM $\phi$ for computing `EAT`, if not provided, we let $\phi \leftarrow \theta$
**Output:** An answer string: A
1: Initialize EMA estimation of mean and variance $\hat{M}_0 \leftarrow 0$, $\hat{V}_0 \leftarrow 0$
2: Initialize reasoning line buffer R $\leftarrow$ [] and line counter $n \leftarrow 0$
3: **while** |R| < $T$ **do**                                          ▷ *|R| denotes the total number of tokens*
4:    Generate new reasoning line $r_n \leftarrow$ GenerateNewLine(Q, <think>, R; $\theta$); ▷*Eq. (3); or use other conditions, e.g., reaching a given token count.*
5:    Append $r_n$ to R, update line counter $n \leftarrow n + 1$
6:    Compute $\text{EAT}_n \leftarrow \mathbb{H}(f(\text{Q}, \text{<think>}, \text{R}, \text{</think>}, \backslash n; \theta))$;                 ▷*Eq. (5)*
7:    Update EMA $\hat{M}_n \leftarrow (1 - \alpha)\hat{M}_{n-1} + \alpha \text{EAT}_n$ , $\hat{V}_n \leftarrow (1 - \alpha)\hat{V}_{n-1} + \alpha(\text{EAT}_n - \hat{M}_n)^2$       ▷*Eq. (7)*
8:    **if** ($n \geq \frac{4}{\alpha}$ **and** $\hat{V}_n < \delta$) **or** </think> is generated **then**       ▷$\frac{4}{\alpha}$ *denotes the EMA warm up iterations.*
9:       **Exit**
10: **Return** A $\sim$ GenerateTillEos(Q, <think>, R, </think>; $\theta$).

---

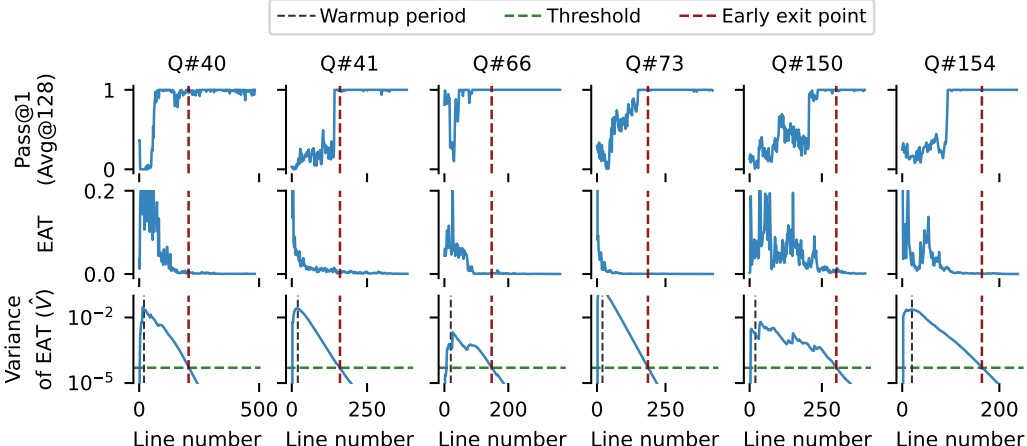

Figure 3: **Illustration of early exiting by thresholding the EMA estimated variance of `EAT`.** We evaluate DeepSeek0528-Qwen8B on various questions from free-form version of GPQA-Diamond (column title denotes question number). As reasoning proceeds, Pass@1 saturates, `EAT` stabilizes, and the variance of `EAT` ($\hat{V}$, Eq. (7), bottom row) decreases. Exiting the reasoning when $\hat{V}$ goes below the threshold (green line) avoids overthinking while maintaining high accuracy.

Lastly, note that Alg. 1 evaluates `EAT` every time "\n\n" is generated. We make this choice mainly for evaluation convenience and consistency with the accuracy metric: We compute Pass@1 (Eq. (8)) at every "\n\n", therefore we use the same schedule for `EAT` to enable direct assessment of early-stopping performance. In practice, other evaluation schedules are also feasible, e.g. every 100 tokens, we provide examples on `EAT` trajectory under alternative frequencies in Appendix E.

### 4.3 `EAT` IS EFFICIENT TO COMPUTE

Recall that our goal is to use `EAT` for early exiting in order for *efficiency*, therefore it is important to understand the overhead of evaluating `EAT`, because at the end of the day, the quantity we care about is not just the number of tokens in the reasoning chain, but the overall wall clock time.

The computation of `EAT` can be conducted very efficiently during the reasoning process: Denote the size of the reasoning chain, measured by the number of tokens, as |R|. Since the KV cache of the reasoning chain is produced during decoding, computing `EAT` only requires one single forward pass, with a deterministic time and memory overhead linear with respect to |R|, roughly equivalent to the overhead of *generating one extra token*. We empirically verify the linear scaling of overhead in Fig. 5c, bottom row: Even with 8K tokens context, the overhead is under 0.1 seconds. This again contrasts `EAT` with rollout-based methods, which must generate multiple hypothetical continuations. Such rollouts introduce stochastic latency (since decoding time depends on the length of sequences), require non-trivial scheduling and parallelization to maximize efficiency, and often need heuristic pruning or beam search style optimization to keep compute manageable.

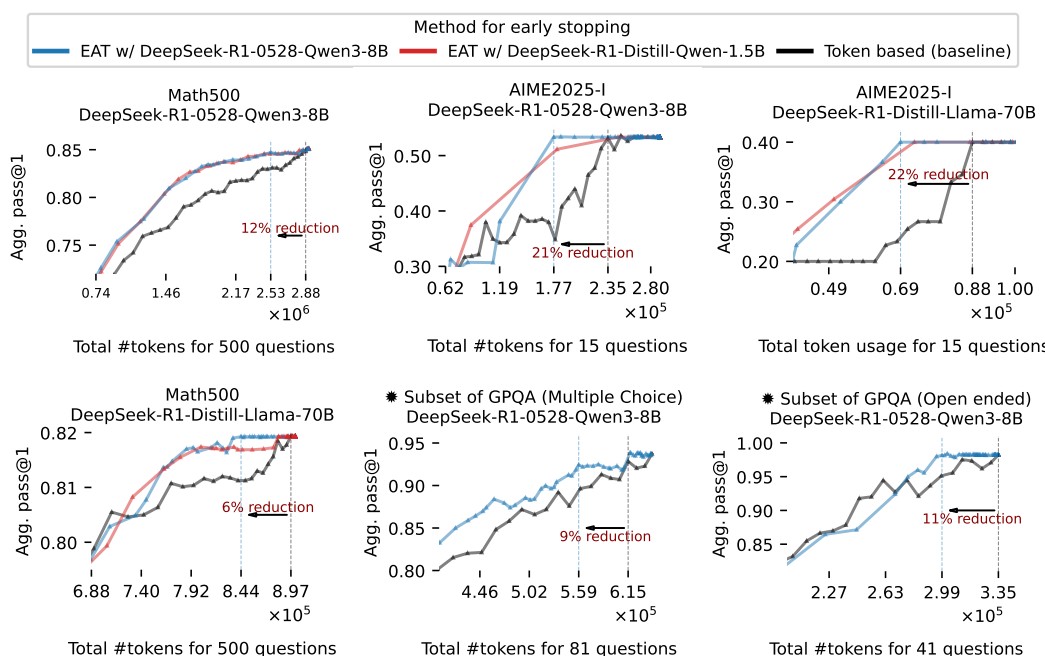

Figure 4: **`EAT`-based early exiting dynamically allocates token budgets and consistently saves tokens without sacrificing accuracy.** Across different datasets and reasoning models (titles show dataset/model), thresholding the variance of `EAT` (blue and red lines, dot denotes a threshold $\delta$ used in Alg. 1) reduces token usage compared to token-based early exiting (black line, dot denotes a fixed per-question token limit $T$), thanks to its adaptivity. Crucially, `EAT` generalizes across model sizes: small proxy models can reliably early-stop much larger reasoning models (e.g., using a 1.5B model to early exit Llama-70B), making the method applicable to black-box APIs.

## 5 EXPERIMENTS: EARLY EXITING WITH `EAT`

In this section, we empirically evaluate `EAT`-based early exiting (Alg. 1). All experiments are conducted on a GPU cluster with 40GB Nvidia A100 GPUs.

### 5.1 EXPERIMENT SETUP

**Models and datasets** We study three reasoning models: DeepSeek-R1-{0528-Qwen3-8B / Distill-Llama-70B}, and Qwen3-4B-Thinking-2507, and two models for computing `EAT`: DeepSeek-R1-{Distill-Qwen-1.5B / 0528-Qwen3-8B}.[3] The datasets for our experiments are Math500 (Lightman et al., 2023), AIME2025, and two versions of GPQA-Diamond (Rein et al., 2024): the standard *multiple-choice* version, where the model needs to pick one option out of four candidates, and the *open-ended* version, where the model needs to come up with its own answer without options provided.

**Output format and correctness check** We prompt the model to put the final answer in `"\boxed{}"` following common practice. For Math500 and AIME2025, correctness is checked by validating the equivalence of equations using SymPy, using the codebase released by Yang et al. (2024). For GPQA Multiple Choice, correctness is determined by exact match with the labeled option. For GPQA open-ended questions, correctness is judged using GPT-4o-mini with a verification prompt from Chandak et al. (2025). All prompts used in experiments are shown in Appendix I.

**Reasoning correctness metric** For each question in the dataset, with question as $Q_d$ and ground true answer as $A_d$, we track Pass@1(Avg@K) after certain reasoning lines $R = r_1, \ldots, r_n$

$$\text{Pass@1(Avg@K)}_d = \frac{1}{K} \sum_{k=1}^{K} \mathbb{1} \left[ A_d^k = A_d \right],$$  (8)

where $A_d^k \sim \text{GenerateTillEoS}(Q_d, \text{<think>}, R, \text{</think>}, \text{Final answer:}\backslash n; \theta)$

---

[3] N.B. we could have used the same models for reasoning as for calculating `EAT`; smaller models are selected here to support a larger number of experiments.

during the reasoning process, where $\mathsf{A}_d^k$ denotes the $k$th randomly sampled rollout from the $\texttt{GenerateTillEoS}(\cdot)$ operation defined in Eq. (3), and the equal sign $\texttt{"="}$ denotes whether $\mathsf{A}_d^k$ is correct verified using techniques in the previous paragraph. Note that $\mathsf{A}_d^k$ does not necessarily always start with $\texttt{"\boxed{}"}$, in practice, it sometimes starts with a summarization of the reasoning process followed by the actual answer. Additionally, the stop thinking token and the $\texttt{"Final}$ $\texttt{answer:\n"}$ strings are *manually added* to force the model to start generating an answer instead of continuing to think after $\texttt{</think>}$. This metric estimates how likely the model is to get the answer correct in one shot, a quantity critical in scenarios where verifiers are not accessible. At the dataset level, we report the aggregated performance. Consider a dataset of $D$ questions, we compute

$$\text{Agg. pass@1} = \frac{1}{D} \sum_{d=1}^{D} \text{Pass@1(Avg@K)}_d. \tag{9}$$

**Implementation details** We discuss additional details on random decoding configurations, LLM inference framework, and efficient implementations of early exiting experiments in Appendix F.

### 5.2 BASELINE METHODS AND EFFICIENCY METRICS

We compare $\texttt{EAT}$-based early exiting with two other early exiting strategies

- **Token-based (Alg. 2)** We set a *fixed* reasoning budget, denoted as $T$, per question. The reasoning stops when either condition is met: i) the stop thinking token $\texttt{</think>}$ is generated; ii) the total reasoning length hits $T$. The advantage is its simplicity and clear physical meaning: The actual overhead is bounded by $\mathcal{O}(D \times T)$. Its drawback lies in the lack of adaptivity: A uniform allocation does not take into account the problem difficulty and the actual progress of the reasoning process.
- **Number of unique answers in K rollouts (#UA@K, Alg. 3)** We use the number of distinct answers in $K$ rollouts answer string as early exiting signal. The second row in Fig. 1 provides an illustration: As reasoning progresses, #UA@128 typically decreases and stabilizes at 1, aligning with Pass@1 convergence. Early exiting can be triggered when #UA@$K$ is smaller than a threshold $\Delta \in \mathbb{N}^+$ or when a token limit $T$ is hit. The advantage of #UA@$K$ lies in its simplicity and adaptivity, as it requires only verbal forms of rollouts, without any model internal information, and the actual token usage changes adaptively depending on the problem. The main drawback of #UA@$K$ is its sensitivity to $K$'s value and the resulting performance-overhead trade-off: A small $K$ introduces too much noise to the signal, whereas a large $K$ causes too much extra overhead.

**Reasoning efficiency metric** Given a dataset, to quantify the goodness of different early exiting approaches, we enumerate over thresholds (the exact values and ranges depend on methods) to construct an Agg. pass@1 (Eq. (9)) v.s. actual total token usage curve. A more efficient early exiting approach should have a larger area under the curve, indicating that fewer tokens are required to achieve a target performance. Note that for all instances in a dataset, we use the same threshold; we believe adaptive thresholds (e.g., different $\delta$s) for each instance can further boost performance.

### 5.3 RESULTS

Now we present the efficiency results of different early exiting methods. For $\texttt{EAT}$ and #UA@$K$, we set the maximum token limit $T$ as $10K$ tokens, and for their thresholds, we sweep $\delta \in 2^{\{0,1,\dots,39\}}$ for $\texttt{EAT}$ and $\Delta \in \{8, 16, 32\}$ and $K \in \{1, 2, 3\}$ for #UA@$K$; For token-based early exiting, we sweep the token limit $T \in 250 \times \{1, 2, \dots, 40\}$. We include a prefix string "$\texttt{Final answer:}$  " (Eq. (11), Appendix B) for all models when computing $\texttt{EAT}$ for better performance.

**Comparison with token-based early exiting** Fig. 4 compares $\texttt{EAT}$ with token-based early exiting (additional results on Qwen-4B are presented in Appendix G.1). Each dot on the lines denotes the Agg. pass@1 and actual total token usage under a specific threshold value, where dots on the black lines denote a value of $T$ for token-based early exiting, and dots on the red and blue lines denote a value of $\delta$ for $\texttt{EAT}$ computed with two different models. For GPQA-Diamond, we filtered the dataset and only kept the problems that the model solved eventually (more details discussed in Appendix G.3). Across datasets and models, $\texttt{EAT}$ consistently reduces token usage (up to 21% on AIME-2025) while maintaining accuracy. Additionally, $\texttt{EAT}$ is effective in black-box setups, where the model for computing $\texttt{EAT}$ differs from the underlying reasoning model. This allows us to use small proxy models (1.5B) to early exit Llama-70B without direct access to its logits. Note that $\texttt{EAT}$ shows less token saving on Llama-70B than Qwen3-8B because Qwen3-8B uses more tokens per question, achieving higher Pass@1 but also overthinking more, leaving greater room for early exit gains.

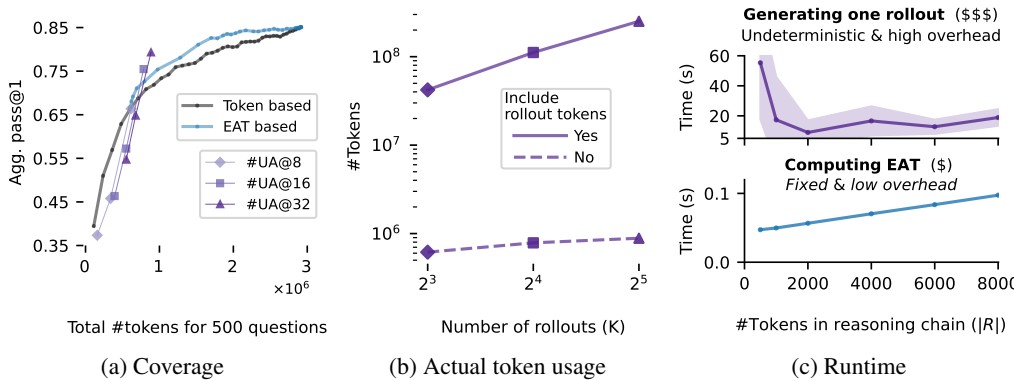

(a) Coverage        (b) Actual token usage        (c) Runtime

Figure 5: **#UA@K shows performance-overhead tradeoff** (a): #UA@$K$ *only* works well when $K \geq 16$ (purple square line); (b): however, if we count the actual token (at $\Delta = 1$) required, including those from the $K$ rollouts, the number is very significant; (c): Generating rollout is expensive even for $K = 1$, and is more than 50 times slower than EAT. The runtime estimation of EAT includes the prefix string "`Final answer:`" and the rollout runtime is estimated with Huggingface implementation.

**Comparison with #UA@K** Fig. 5 compares EAT with #UA@$K$ on DeepSeek0528-Qwen8B under Math500. Each purple line denotes a specific value of $K$ and each dot denotes a value of unique answer number thresholds $\Delta$. The metric value at each dot is the average over 64 random samples. The results show several critical issues with #UA@$K$. Firstly, the performance is sensitive to $K$: When $K$ is small (lightest diamond purple line in Fig. 5a), the estimation is inaccurate (since #UA@$K$ is upper bounded by $K$) so #UA@$K$ underestimates the actual value therefore stopping too early and underperforms other methods. When $K$ is large (darkest triangle purple line in Fig. 5a), we can achieve very good early exiting performance; however, the overhead will be very significant, which goes *against* the motivation of early exiting. The reason is that rollouts are expensive: generating them dominates both token usage (Fig. 5b) and runtime (Fig. 5c top) in a stochastic way. In contrast, EAT is deterministic, cheap to compute, and stable across runs.

**Ablation study on EAT's hyperparameter** We provide ablation studies on the effect of EMA timescale $\alpha$ and the effect of prefix string for early exiting performance in Appendix G.2, where we find that EAT remains effective as long as $\alpha > 0.1$, with or without the prefix string.

**Error analysis of EAT** For GPQA, we only present early exiting results on solvable subsets in Fig. 4, where we only kept problems for which the models eventually reached a Pass@$1 > 0.8$. The reason is that there are many problems the model cannot solve and EAT will use up all $T$ tokens due to the $\hat{V}$ never going below $\delta$, we present examples and more discussions in Appendix G.3.

## 6 CONCLUSIONS AND FUTURE WORK

In this paper, we propose a simple metric to monitor the reasoning process. In particular, we look at EAT, the entropy over the distribution of the next token after `</think>`. We find this quantity informative for early exiting the reasoning process and extremely efficient to compute. Notice that while current reasoning models can exhibit severe overthinking, it is certainly possible that future reasoning LLMs will use improved post-trained algorithms (Qi et al., 2025; Dai et al., 2025; Shrivastava et al., 2025) under which the model can automatically stop reasoning when the performance improvement has saturated, in which case early exiting becomes less effective. Still, we believe EAT quantity can serve as a metric that monitors and diagnoses the reasoning process, similar to the roles of, e.g., auto-correlation in Markov chain Monte Carlo (Gelman et al., 1995; Vehtari et al., 2021), gradient norm in stochastic optimization, model FLOPs utilization (Pope et al., 2023).

There are several avenues for future work. We here treated problems as coming from a common distribution, analyzing early exit per instance with a single global threshold and without cross-problem budget allocation. Future work could relax these assumptions by adapting thresholds for each problem or jointly optimizing budget allocation. We also focused here on short-circuiting reasoning that eventually stabilizes, a promising extension is to learn to recognize and abandon reasoning that will never stabilize, e.g., considering exiting when EAT or its variance remains high for many steps.

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

## A  LLM USAGE DISCLOSURE

Large language models are only moderately used for editing and improving the coherence of the text; they are not used to provide research or writing ideas in any form.

## B  PREFIX STRING FOR COMPUTING EAT

In the main text, the EAT trajectories we illustrate in Fig. 1, 2 and 3 are computed as

$$\text{EAT} = \mathbb{H}(f(\text{Q}, \texttt{<think>}, \text{r}_1, \ldots, \text{r}_n, \texttt{</think>}, \backslash \text{n}; \theta)), \tag{10}$$

using the latest model DeepSeek0528-Qwen8B (released May 2025), where we observe that the stabilization of EAT correlates with the saturation of Pass@1.

For other models, such as DeepSeek-R1-Distill-Qwen-1.5B and DeepSeek-R1-Distill-Llama-8B (both released Feb 2025), we notice that a prefix string needs to be appended in order for EAT to be informative. In particular, we refer to the following quantity

$$\text{EAT}_{\texttt{prefix}} = \mathbb{H}(f(\text{Q}, \texttt{<think>}, \text{r}_1, \ldots, \text{r}_n, \texttt{</think>}, \backslash \text{n}, \texttt{The final answer: } ; \theta)), \tag{11}$$

which includes "The final answer:" followed by an empty space after the `</think>`\n.

We present some illustrations of EAT computed with and without the prefix under various models in Fig. 6a and 6b. Given reasoning trajectories from DeepSeek-R1-{0528-Qwen3-8B / Distill-Llama-70B}, we notice that $\text{EAT}_{\texttt{prefix}}$ is necessary for the old models (models released in late Jan), but not necessary for new models such as DeepSeek0528-Qwen8B and Qwen3-4B-Thinking-2507.

In practice, we find that using $\text{EAT}_{\texttt{prefix}}$ works better for early exiting performance, as it correlates more with Pass@1 compared with EAT, detailed results are presented in Appendix G.2. It is also worth noting that the memory and time overhead introduced by the extra prefix string is negligible in that all tokens inside the prefix string can be prefilled in parallel; therefore, the overhead will still be dominated by the length of the reasoning tokens.

## C  EARLY EXITING ALGORITHM

---
**Algorithm 2** Token-based early exiting

---
**Input:** Question Q, max token limit $T$;
**Input:** Reasoning LLM $\theta$
**Output:** An answer string: A
1: Initialize reasoning line buffer R $\leftarrow$ [] and line counter $n \leftarrow 0$.
2: **while** $|\text{R}| < T$ **do**                                         ▷$|R|$ *denotes the total number of tokens*
3:     Generate new reasoning line $\text{r}_n \leftarrow$ GenerateNewLine(Q, `<think>`, R; $\theta$);          ▷*Eq.* (3)
4:     Append $\text{r}_n$ to R, update line counter $n \leftarrow n + 1$
5:     **if** `</think>` is generated **then**
6:         **Exit**
7: **Return** A $\sim$ GenerateTillEos(Q, `<think>`, R, `</think>`; $\theta$).

---

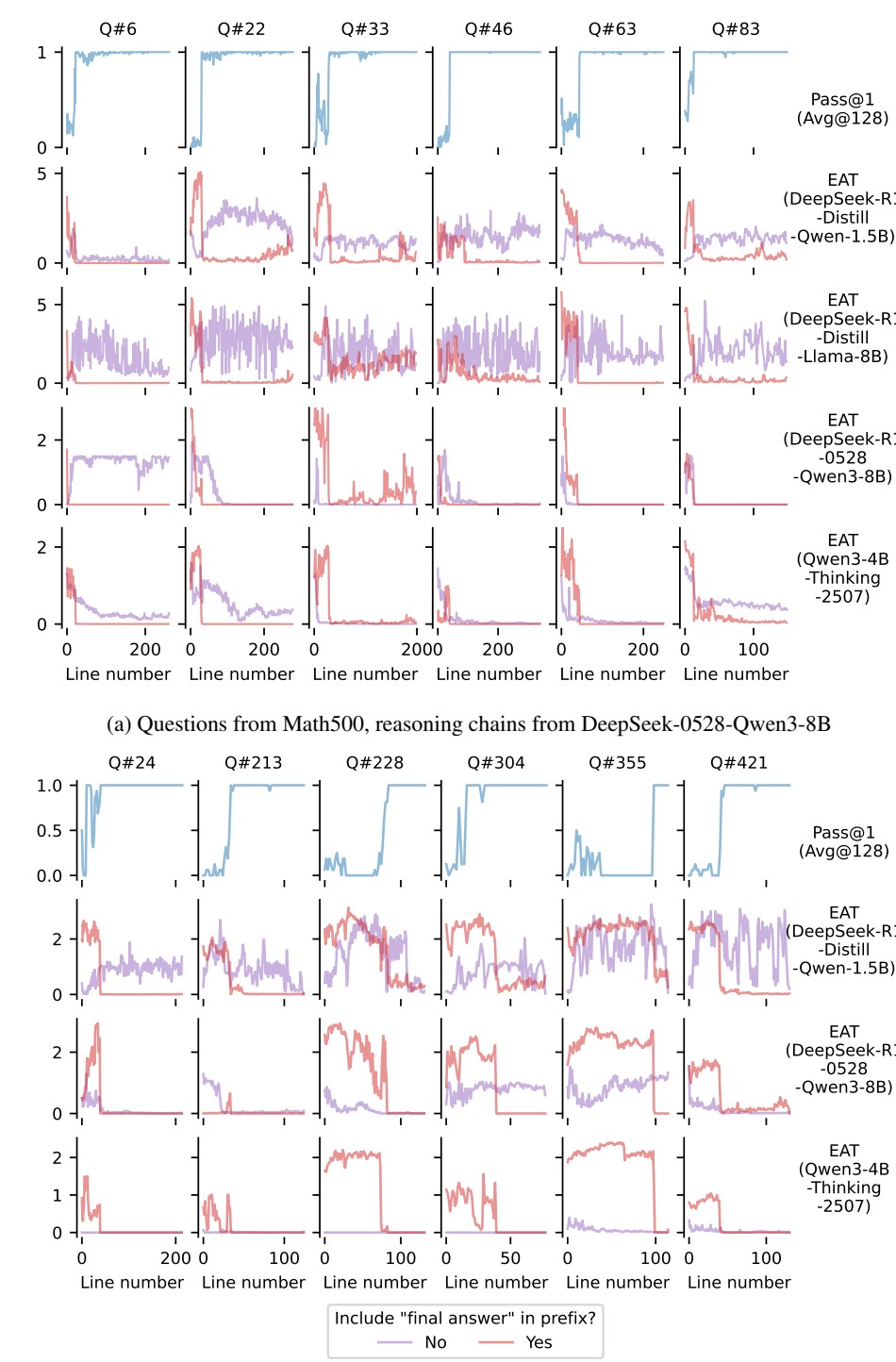

(a) Questions from Math500, reasoning chains from DeepSeek-0528-Qwen3-8B

(b) Questions from Math500, reasoning chains from DeepSeek-R1-Distill-Llama-70B

Figure 6: **EAT trajectories computed with and without a prefix string "The final answer:" for different reasoning models and Math500 questions.** )(a) Reasoning chains from DeepSeek-0528-Qwen3-8B: older proxy models (DeepSeek-R1-Distill-Qwen-1.5B and DeepSeek-R1-Distill-Llama-8B) require adding the prefix (red lines) for EAT to align with Pass@1 saturation, whereas the newer DeepSeek-0528-Qwen3-8B and Qwen3-4B-Thinking-2507 do not (purple lines). (b) Reasoning chains from DeepSeek-R1-Distill-Llama-70B show the same trend. In all cases, adding the prefix introduces negligible overhead because its tokens can be prefilled in parallel.

---

**Algorithm 3** #UA@$K$ based early exiting

---

**Input:** Question Q, token limit $T$; Number of rollouts $K$, unique answer threshold $\Delta$
**Input:** Reasoning LLM $\theta$
**Output:** An answer string: A
1: Initialize reasoning line buffer R $\leftarrow$ [] and line counter $n \leftarrow 0$.
2: **while** |R| < $T$ **do**                 ▷|R| *denotes the total number of tokens*
3:     Generate new reasoning line $r_n$ $\leftarrow$ GenerateNewLine(Q, <think>, R; $\theta$);     ▷*Eq. (3)*
4:     Append $r_n$ to R, update line counter $n \leftarrow n + 1$
5:     Randomly generate $K$ answer rollouts
    A$^k$ $\sim$ GenerateTillEoS(Q,<think>, R, </think>, Final answer:\n;$\theta$), $k \in 1, \ldots, K$
6:     Extract and *count* all unique answers from A$^1$, ..., A$^K$, denoted as $U$
7:     **if** ($U \leq \Delta$) **or** </think> is generated **then**
8:         **Exit**
9: **Return** A $\sim$ GenerateTillEos(Q, <think>, R, </think>; $\theta$).

---

## D    COMPARISON WITH ENTROPY AFTER NEW LINE

Our empirical evidence shows that the entropy after </think> correlates with Pass@1 and the predictive distribution after. We did not find the entropy inside the chain of thought to be informative. In particular, we look at

$$\text{Entropy after newline} = \mathbb{H}(f(\text{Q}, \texttt{<think>}, r_1, \ldots, r_n, \text{\textbackslash n\textbackslash n}; \theta)). \tag{12}$$

Examples are shown in the last row in Fig. 7.

We suspect that the entropy after newline may correlate with the internal stage of reasoning, e.g. whether the model (Wang et al., 2025) is in the middle of reasoning step or is about to finish and start a new round of revision, but less correlated with whether a confident answer is conditioned on the reasoning, a quantity Pass@1 reflects.

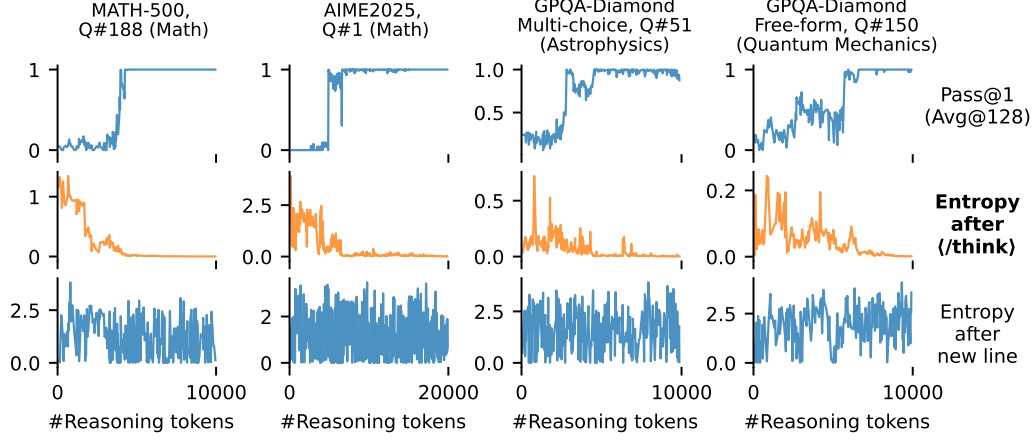

Figure 7: Similar to Fig. 1, but we include Entropy after newline (last row), which has the same overhead as EAT but is less informative and correlated with Pass@1.

# E    EVALUATING EAT AT ALTERNATIVE FREQUENCIES

In the main text, we compute EAT every time "\n\n" is generated; the main reason is that for the reasoning models we considered, the lengths distribute fairly equally across different paragraphs and we choose to compute the expensive Pass@1 (Avg@128) at every "\n\n" for implementation convenience. However, one may raise concerns that not all reasoning models output "\n\n", certain reasoning models could generate the whole CoT in one single long sentence (although it is unclear which model behaves like this).

Under such scenarios, one can also evaluate EAT at alternative scheduling, e.g., at every $S$-token. We provide illustrations of EAT trajectory evaluated every $S$-token under $S \in \{50, 100, 200\}$ in Fig. 8, where we find that the overall pattern of EAT stays unchanged, and EAT still correlates with the Pass@1 trajectory. However, it is *not* feasible to *evaluate* the actual early exiting performance under token-based scheduling as each dot on the EAT line no longer aligns with the points on the Pass@1 trajectory.

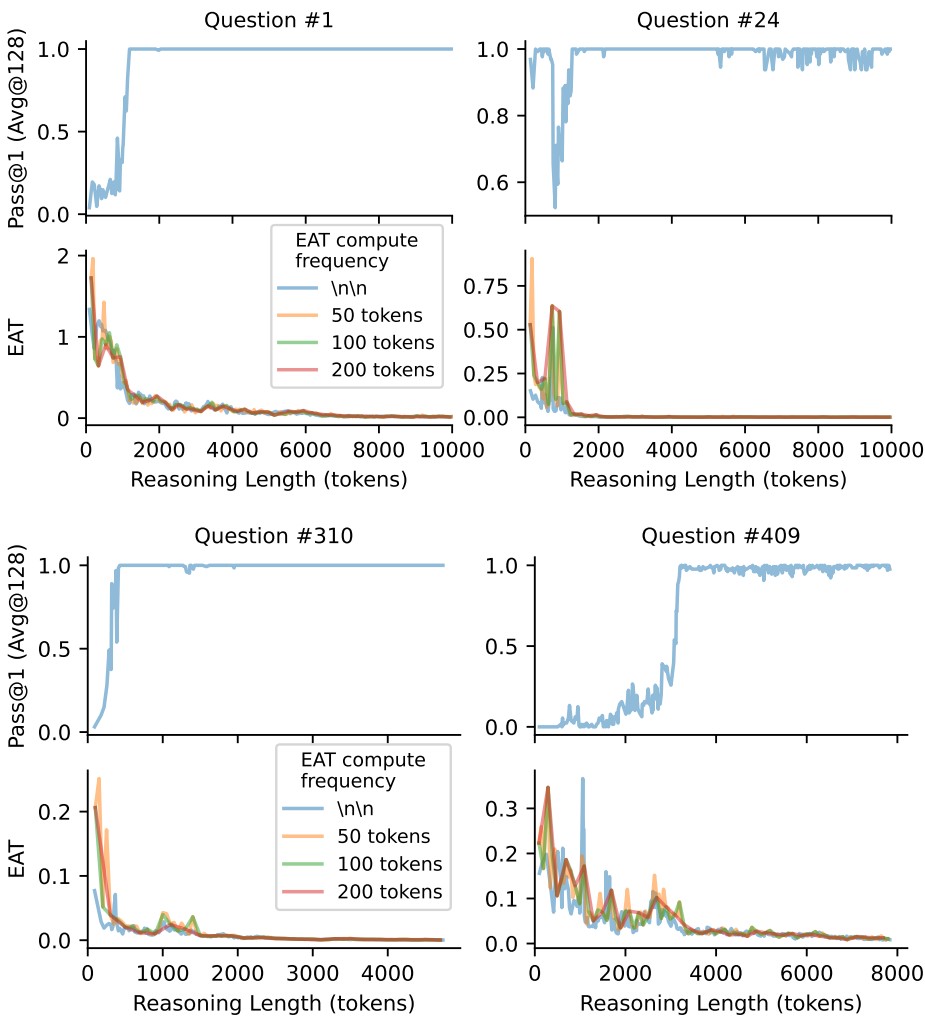

Figure 8: **EAT computed under different frequencies shows patterns.** Here we evaluate DeepSeek-R1-0528-Qwen3-8B on questions from Math500. Given the same reasoning trajectory, we evaluate EAT at different frequencies: Every new paragraph (blue line) and every $S$ tokens (the rest of the lines). The overall behavior of EAT stays unchanged, except that the trajectory becomes smoother at a higher value of $S$.

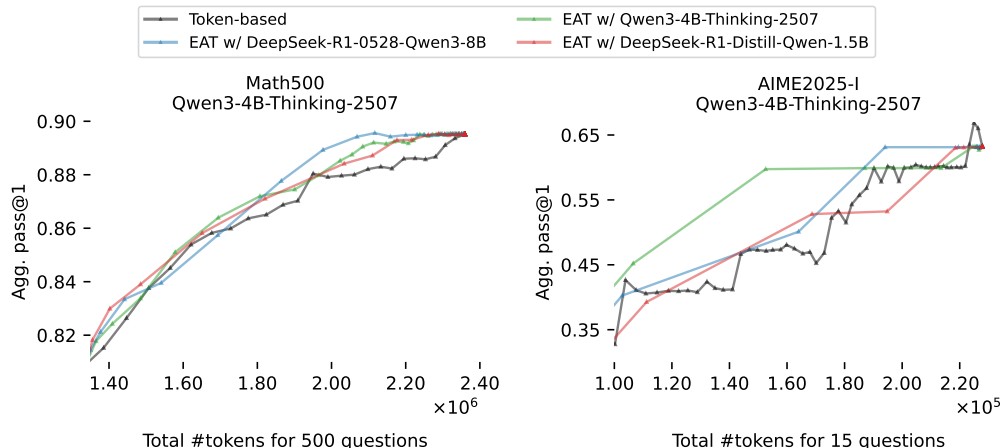

Figure 9: Performance of `EAT`-based early stopping with Qwen3-4B-Thinking-2507 as the reasoning model, experiment setting similar to Fig. 4 in the main text. Each dot on the black line denotes a per-question token limit threshold $T$ and each dot on the other lines denotes a variance threshold $\delta$. `EAT` saves tokens due to its adaptivity. Additionally, one can perform `EAT`-based early exiting using either the original reasoning model (green line) or other reasoning models (blue and red lines).

## F  ADDITIONAL EXPERIMENT SETTING

**Implementation details**  In our experiments, we generate the reasoning chain and compute `EAT` using the native Huggingface implementation. When generating answer rollouts, i.e. the one used to compute Eq. (8) and for estimating #UA@$K$, we use vLLM (Kwon et al., 2023) for efficiency. For the generation of both the reasoning chains and the rollouts, we use a decoding configuration with temperature 0.6 and top-p of 0.95, following the official recommendations in the model card [4]. Due to resource limitations, for each question in each dataset, only one random reasoning chain is generated for evaluating early exiting.

**Simulated early exiting**  When conducting experiments, we perform early exit evaluation in a post-hoc manner. We first generate a single, very long chain of thought for each question, storing the complete reasoning trace and computing the Pass@1 and `EAT` trace. We then simulate early exiting by *retrospectively* truncating this chain at different stopping points. As such, for every instance in the dataset, we can analyze test time scaling and early exit behavior simply by producing one long reasoning chain, saving it once to disk, and replaying it offline to compute metrics at arbitrary exit thresholds without re-querying the model.

## G  ADDITIONAL EXPERIMENT RESULTS

### G.1  EXPERIMENT RESULTS ON QWEN4B

Fig. 9 presents comparisons between `EAT`-based vs. token-based early exiting using Qwen3-4B-Thinking-2507 as the reasoning model evaluated on Math500 and AIME2025, and we compute `EAT` using three different models: Qwen3-4B-Thinking-2507 (green lines), DeepSeek-R1-Distill-Qwen-1.5B (red lines) and DeepSeek-R1-0528-Qwen3-8B (blue lines). The observation aligns with that in Fig. 4 in the main text, where `EAT` shows more efficient usage of tokens and remains effective when the model computing `EAT` differs from the reasoning model.

### G.2  ABLATION STUDY

We conduct ablation studies to understand how the exponential moving average (EMA) timescale $\alpha$ and the optional "Final answer: " prefix influence the performance of `EAT`-based early exiting.

For a fixed reasoning dataset (Math500), we sweep $\alpha$ over $\{0.01, 0.05, 0.1, 0.2, 0.4\}$. For each $\alpha$, we measure the area under the Pass@1–token-usage curve (AUC), which quantifies how efficiently

---

[4] https://huggingface.co/deepseek-ai/DeepSeek-R1-0528-Qwen3-8B
https://huggingface.co/deepseek-ai/DeepSeek-R1-Distill-Llama-70B

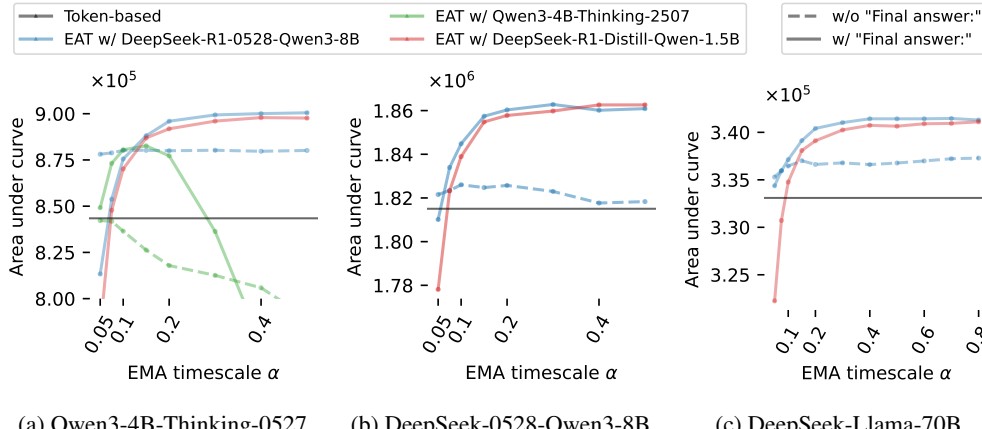

(a) Qwen3-4B-Thinking-0527    (b) DeepSeek-0528-Qwen3-8B    (c) DeepSeek-Llama-70B

Figure 10: **EAT outperforms token-based early stopping with or without prefix string included, under various values of EMA timescale $\alpha$.** Evaluated by the area under Agg. pass@1 v.s. total token usage curve (y-axis), where a larger value implies more efficient usage of tokens, we study three different reasoning models' performance on Math500 (subfigures), under various configurations of computing `EAT` (different lines) and values of timescale $\alpha$ (x-axis). Broadly, we find that a short time window, i.e. $\alpha$ value of about 0.1-0.2, works well for most methods.

a method maintains accuracy while saving tokens. We repeat the procedure both with and without appending the "Final answer: " prefix string when computing `EAT`.

Fig. 10 reports the resulting AUC as a function of $\alpha$ for Qwen3-4B-Thinking-2507, DeepSeek-R1-0528-Qwen3-8B and DeepSeek-R1-Distill-Llama-70B. `EAT` remains effective over a broad range of $\alpha$ values as long as $\alpha > 0.1$. Extremely small $\alpha$ (e.g., 0.01) averages over too long a window, blurring short-term fluctuations and causing delayed stopping, which lowers the AUC. In contrast, moderate $\alpha$ $(0.1 - 0.4)$ balances responsiveness with stability, yielding consistently high AUC across both models. Including "Final answer: " in the prompt slightly AUC across all $\alpha$, which confirms that the prefix tightens the coupling between `EAT` stabilization and Pass@1 convergence.

### G.3 GPQA-DIAMOND FILTERING AND ERROR ANALYSIS

In the main text, when evaluating `EAT`'s early exiting performance on both versions of GPQA-Diamond ( Fig. 2, last two columns, bottom row), we kept only samples where Pass@1(Avg@128) reaches 0.8 at the end of reasoning. The reason is that many instances in GPQA-Diamond are either not solvable or the model would show decreasing Pass@1 throughout reasoning. For such questions, the naive token-based early exiting would be preferable to `EAT`. We present detailed discussion and examples of these two cases below, using questions from open-ended version of GPQA-Diamond.

**Non-solvable question** Due to the difficulty of GPQA-Diamond, the reasoning model often completely fails to solve some of the questions, regardless of the reasoning efforts. Examples are shown in Fig. 11, where the Pass@1 stays constantly low throughout the reasoning. In such scenarios, `EAT` does not stabilize; therefore Alg. 1 would use up all $T$ tokens.

**Questions with decreasing Pass@1** Certain problems begin with a relatively high Pass@1, but the value gradually declines as reasoning continues (Fig. 12). Here `EAT` often fails to stop at the optimal exit position, which does not show an obvious correlation with `EAT`.

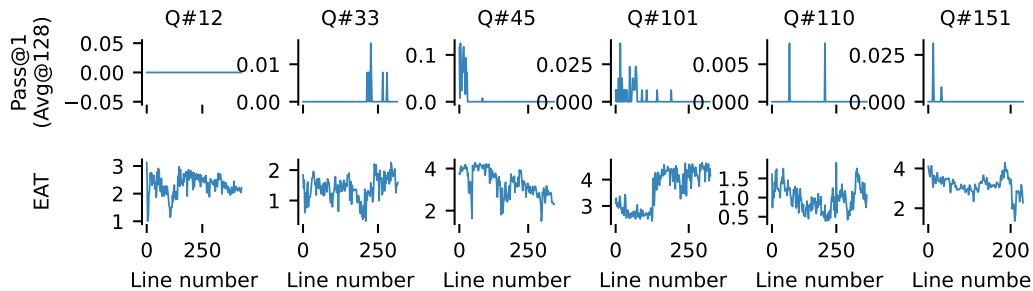

Figure 11: On unsolvable question, `EAT` does not stabilize and therefore would use up all tokens under Alg. 1.

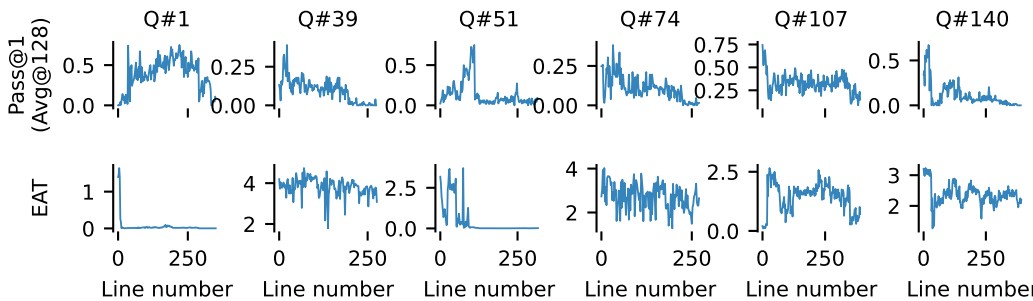

Figure 12: On questions with decreasing Pass@1, `EAT` either does not stabilize or stabilizes at positions where the optimal Pass@1 position has passed.

### G.4 COMPARISON WITH ONE-PASS CONFIDENCE

Here we compare `EAT` with the confidence score proposed by (Yang et al., 2025). In particular, given a reasoning trace R, Yang et al. (2025) proposes to roll out $T$ tokens (denoted as $\{a_1, \ldots, a_T\}$) *greedily* after R appended with an answer-inducing string (`"</think>, Final answer:\n"`), and computed the confidence as

$$\text{Confidence}(R) = \exp\left(\frac{1}{T}\sum_{t=1}^{T}\log p(a_t \mid R, \{a_{t'}\}_{t'<t})\right) \tag{13}$$

i.e. the length-normalized likelihood for a single generated rollout. Note that although not discussed in the original paper, the confidence score is equivalent to a single-sample estimation of the $T$-gram entropy over $\{a_1, \ldots, a_T\}$ if we transform it back in the log space and remove the length normalization.

We provide some examples of confidence vs. `EAT` in Figure. 13. Overall, we find that `EAT` shows similiar correlation with Pass@1 compared with confidence, while confidence is slightly noisier as it is a quantity over multiple tokens.

We include the early stopping performance comparison in Fig. 14, where we evaluate `EAT` and confidence (with $T = 5$) at the same position in the reasoning chain, and threshold the EMA estimated variance as early stopping criteria. We find that EAT performs as well as the confidence score for early stopping, despite being 5 times cheaper.

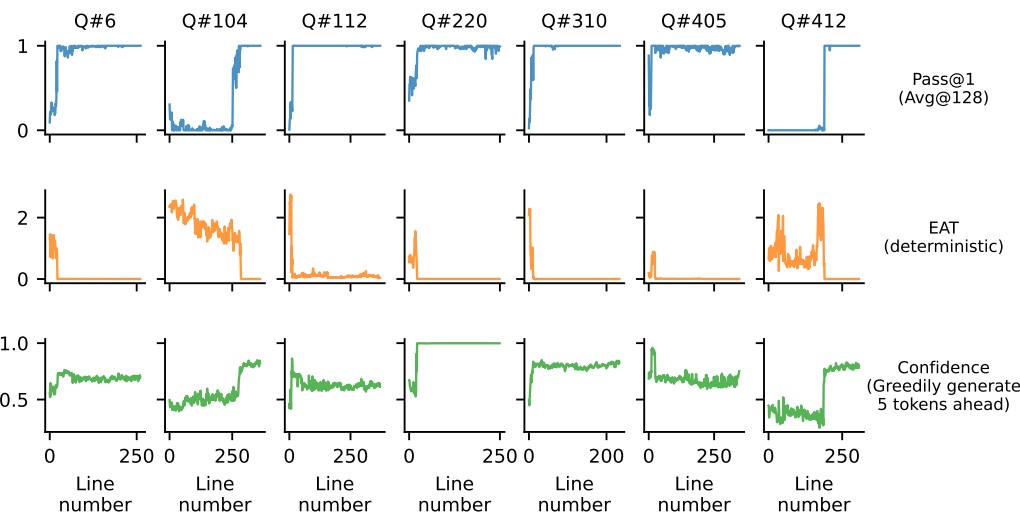

Figure 13: Both `EAT` (Eq. (5)) and confidence (Eq. (13)) stabilizes as Pass@1 plateaus.

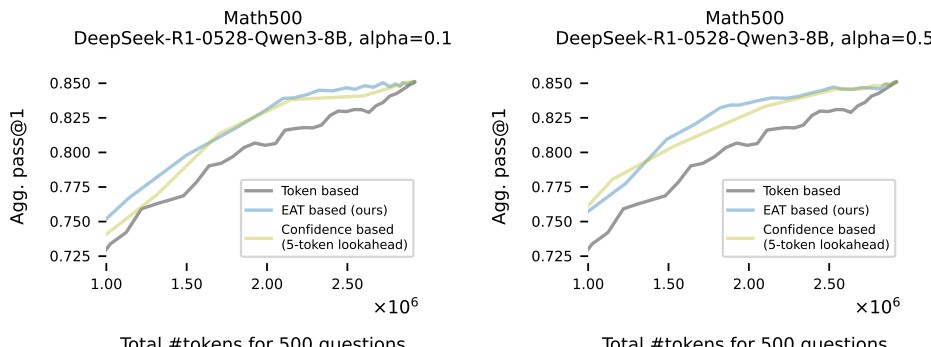

Figure 14: `EAT` shows similiar early stopping performance compared with confidence (Eq. (13), (Yang et al., 2025)), at two different EMA windows sizes $\alpha$, while being 5 times cheaper.

### G.5 MATH500 ERROR ANALYSIS

We present examples where the reasoning model fails to solve at the end of reasoning (10,000 tokens) in Fig. 15.

### G.6 EARLY STOPPING FOR CLAUDE 3.7 SONNET

We demonstrate using a locally deployed Qwen3-4B-Thinking-2507 to compute EAT for the reasoning trace received from Claude 3.7 (provided by OpenRouter), an API model, evaluated on the first 8 questions of AIME2025.

Note that the API returns the thinking tokens block by block in a streaming way (approximately 5 tokens per block), and we compute `EAT` every time we receive a chunk of 20 blocks.

We show the trace of Pass@1 (Avg@1), `EAT`, and elapsed time in Figure. 16 v.s. chunk index, where the gray dashed line denotes the early stopping point detected by our algorithm, under $\alpha = 0.2$ and $\delta = 10^{-3}$. The results indicate that our algorithms, on solvable problems, save *at least one minute* for each query.

Importantly, the computation of `EAT` can overlap with the generation of reasoning tokens, that is, we compute `EAT` for the most recently received chunk while we are waiting for the next chunk. The feasibility of the overlapped compute is shown in Figure 17, where we verify that the wallclock time

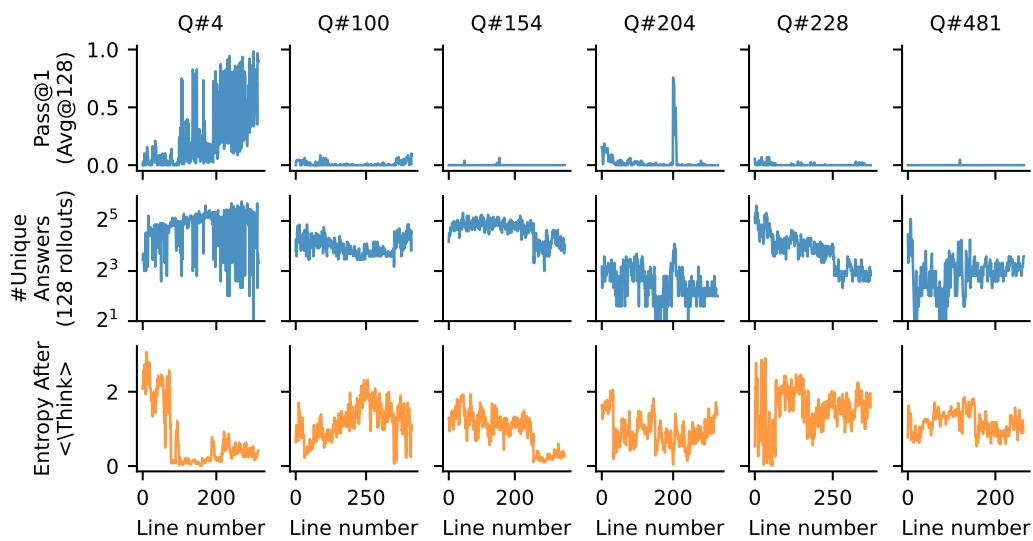

Figure 15: On questions from Math500 with low Pass@1 (Avg@128), evaluated on DeepSeek-R1-0528-Qwen3-8B, the number of unique answers as well as `EAT` stays at a high value and perturbs significantly, indicating that the model is struggling to reach a certain and confident answer.

---

**Algorithm 4** Warmup free early exiting reasoning process based on variance of `EAT`.

---

**Input:** Question Q, max token limit $T$; Variance threshold $\delta > 0$, EMA timescale $\alpha \in (0, 1)$.
**Input:** Reasoning LLM $\theta$; (*Optional*) A proxy LLM $\phi$ for computing `EAT`, if not provided, we let $\phi \leftarrow \theta$
**Output:** An answer string: A
 1: Initialize EMA estimation of mean and variance $\hat{M}_0 \leftarrow 0, \hat{V}_0 \leftarrow 0$
 2: Initialize reasoning line buffer R $\leftarrow$ [] and line counter $n \leftarrow 0$
 3: **while** |R| $< T$ **do**              $\triangleright$|R| *denotes the total number of tokens*
 4:  Generate new reasoning line $\text{r}_n \leftarrow$ GenerateNewLine(Q, <think>, R; $\theta$); $\triangleright$*Eq. (3); or use other conditions, e.g., reaching a given token count.*
 5:  Append $\text{r}_n$ to R, update line counter $n \leftarrow n + 1$
 6:  Compute $\text{EAT}_n \leftarrow \mathbb{H}(f(\text{Q},\text{<think>},\text{R},\text{</think>},\text{\textbackslash n}; \theta));$        $\triangleright$*Eq. (5)*
 7:  Update EMA $\hat{M}_n \leftarrow (1 - \alpha)\hat{M}_{n-1} + \alpha\text{EAT}_n$ , $\hat{V}_n \leftarrow (1 - \alpha)\hat{V}_{n-1} + \alpha(\text{EAT}_n - \hat{M}_n)^2$  $\triangleright$*Eq. (7)*
 8:  $\hat{V}_n \leftarrow \frac{\hat{V}_n}{1-(1-\alpha)^n}$                      $\triangleright$*De-biasing*
 9:  **if** $(\hat{V}_n < \delta)$ **or** </think> is generated **then**
10:   **Exit**
11: **Return** A $\sim$ GenerateTillEos(Q, <think>, R, </think>; $\theta$).

---

for computing `EAT` is much smaller than the wait time of receiving the reasoning chunk from the API.

### G.7 WARM-UP FREE `EAT`-BASED EARLY STOPPING

In the main text, we included a warm-up period of $4/\alpha$ steps in Alg. 1 to alleviate the bias from the initial value of $\hat{V}_0 = 0$, which would cause underestimation of the variance in the early phase of the reasoning. An alternative approach to alleviate this bias without using warm-up phase is to scale the EMA estimation by $\frac{1}{1-(1-\alpha)^n}$, a factor that will gradually decay to 1 as we observe more samples.

The performance of the warm-up free version is presented in Figure. 18

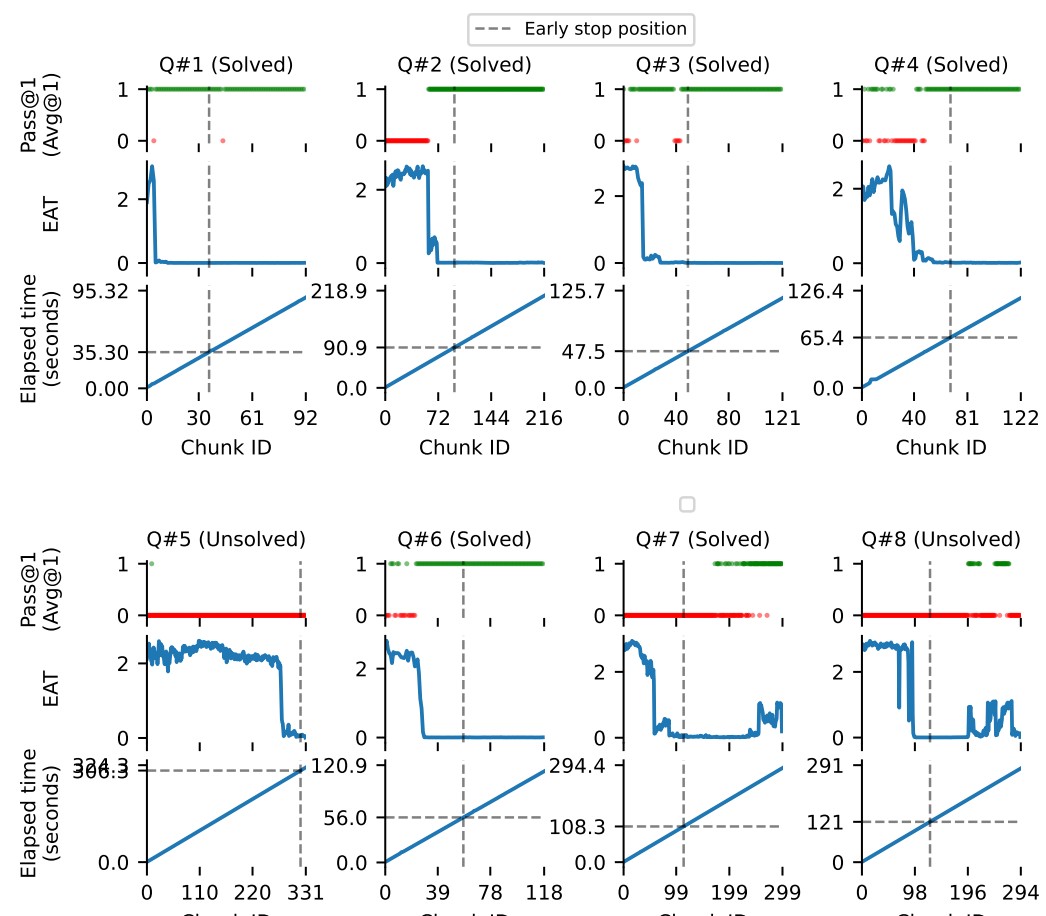

Figure 16: On the first 8 questions from AIME2025, we generate reasoning using Claude 3.7 and compute `EAT` with Qwen3-4B-Thinking-2507. The gray dashed line indicates the early stopping position acquired by thresholding the variance at $10^{-3}$. The in brackets (solved / unsolved) denotes the correctness of the final answer included in Claude's auto summarization of the whole reasoning. Our approach successfully early stops all solvable problems except for Q#7, saving at least 60 seconds for each question.

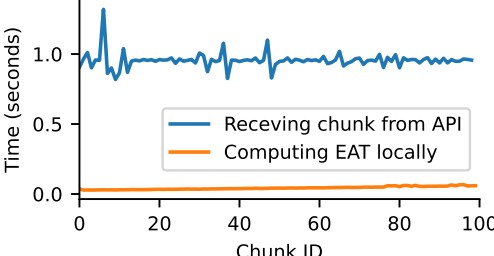

Figure 17: Computing EAT using a local model, time measured on Nvidia L40s averaged over 10 independent runs. The time to compute EAT when receiving a new chunk of tokens is always much smaller than the time of receiving tokens from the server, therefore, we can always do overlapped computing, where the main process performing API call does not need to wait.

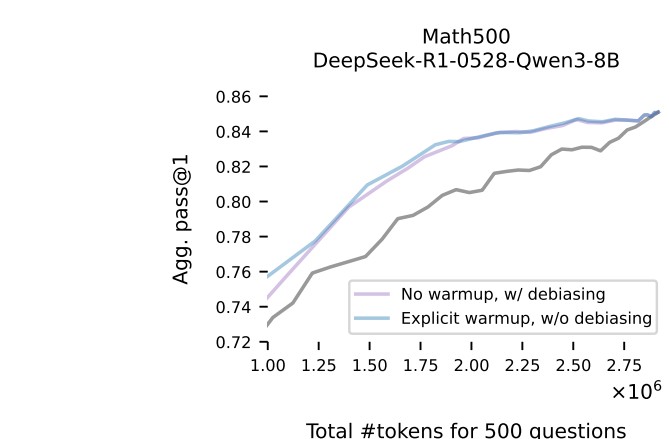

Figure 18: Performance of standard (Alg. 1) vs. debiased version (Alg. 4) of `EAT`-based early exiting. Removing the warm up phase does not affect the performance.

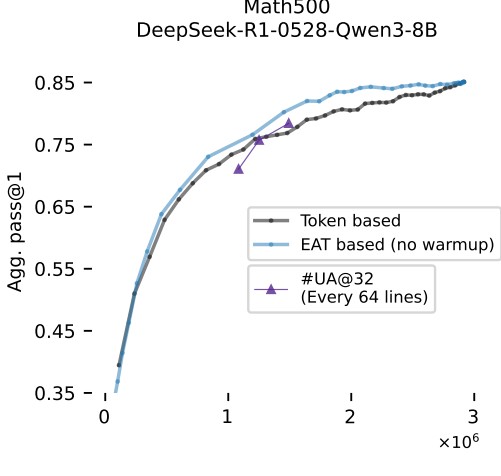

Figure 19: Comparision between #UA@32 v.s. `EAT` under similiar computing budget, where we evaluate #UA@32 every 64 lines to compensate for the overhead of generating 32 rollouts.

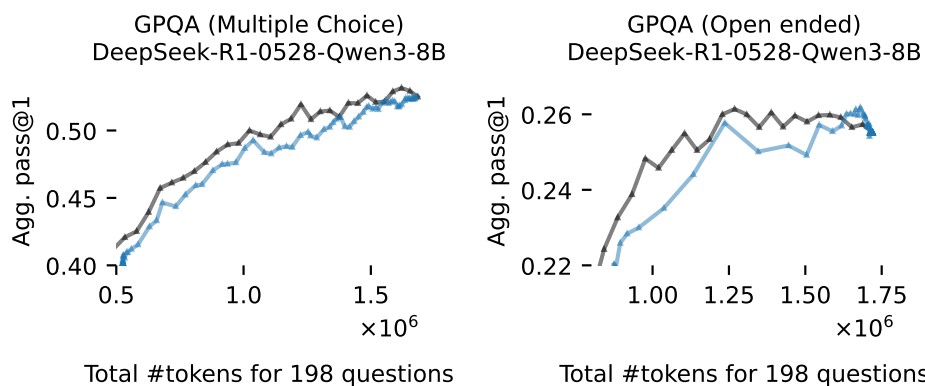

Figure 20: Performance of `EAT` based early stopping on unfiltered GPQA-Diamond.

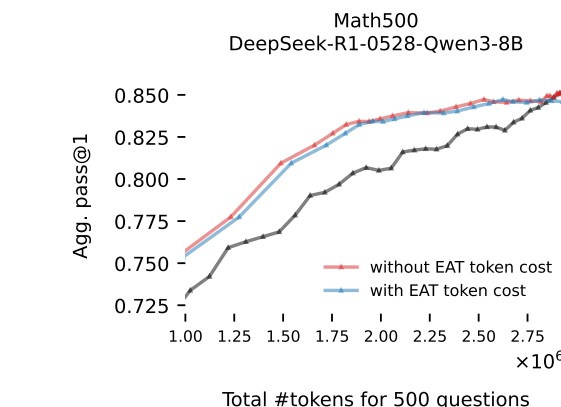

Figure 21: Performance v.s. total token usage, where we additionally take into account the overhead introduced by computing `EAT` (counted as 1 token for every evaluation). `EAT` based early stopping still shows efficiency gains, thanks to the minimal overhead of evaluation.

## H    EXAMPLE OF REASONING MODEL OUTPUTS

We query one of the newest reasoning models (DeepSeek-R1-0528-Qwen3-8B) with a simple question: "What are the first seven digits of Pi?". The code block below shows the model's output under the recommended decoding strategy, random decoding with a temperature of $0.6$ and a top-p value of $0.95$. The bold black text shows the question body, and the bold red text denotes the special begin-of/end-of thinking tokens.

The model arrives at the correct answer in its very first paragraph, yet continues to recheck and restate the same conclusion across more than ten subsequent paragraphs without any meaningful revision, which we consider as qualitative evidence of overthinking.

```
What are the first seven digits of Pi?<think>
Okay, the user is asking for the first seven digits of Pi. Let me start
by recalling what Pi is. Pi is the ratio of a circle's circumference to
its diameter, approximately 3.1415926535... So the first seven digits
would be 3.141592. Wait, I need to count carefully. Starting from 3, then
 1,4,1,5,9,2. That's seven digits after the decimal, but including the
whole number part, it's 3.141592, which is eight digits total. Hmm, maybe
 the user means the first seven digits after the decimal? Let me check
the question again. They said "first seven digits of Pi," which could be
ambiguous. Commonly, people refer to the digits after the decimal as the
"digits of Pi." But sometimes, the whole number is included. I should
clarify this to avoid confusion.

Wait, the user might be a student or someone just starting to learn about
 Pi. They could be working on a project or homework that requires the
initial digits. Maybe they need it for a calculation, programming, or a
school assignment. Including the whole number part would be
3.14159265..., so the first seven digits including the 3 would be up to
3.141592. But the question specifically says "digits," not "digits after
the decimal." I should present both possibilities but clarify the
standard interpretation.

Also, the user might not know that Pi is an irrational number with
infinite digits. They might just need the first seven for a specific
purpose. Are they calculating something that requires a certain precision
? Or maybe they're curious about Pi's properties. Since the answer is
straightforward, I can provide the digits after the decimal and mention
the whole number part if needed. Let me make sure the response is clear
and concise, avoiding any technical jargon unless necessary.
```

Including the 3 would be the first seven digits as a whole: 3,1,4,1,5,9,2. Without the 3, it's seven digits after the decimal: 1415926. But the common way to refer to the first seven digits is usually including the 3. Wait, no. Let me think. If someone says "the first seven digits of Pi," they often mean the initial seven digits, which includes the 3. For example, Pi starts with 3.141592..., so the first seven digits are 3.141592. But the user might be expecting the digits after the decimal.

I should provide both answers but specify which is which. The user's main need is the first seven digits, so I'll present the sequence including the 3 and then the seven digits after the decimal. That way, it covers both interpretations. Also, maybe the user is testing if I know the exact digits or if there's a trick. But since Pi is well-known, the answer should be straightforward.

Let me verify the digits. Pi is approximately 3.1415926535, so the first seven digits are 3.141592. If they want seven digits after the decimal, it's 1415926. But the question doesn't specify, so I'll go with the common understanding. The user might not have specified, but in most contexts, "digits of Pi" includes the integer part.

I'll structure the answer by first giving the seven digits including the 3, then clarifying the seven digits after the decimal. That way, the user gets the information they need without confusion. Make sure to explain briefly so they understand the difference.

Also, considering the user's possible deeper needs: they might be preparing for a presentation, writing a program that needs Pi's digits, or just curious. By providing both, I address different possible use cases. They might not have realized Pi includes the 3, so this could help them in the future.

I should mention that Pi is irrational and non-repeating, but that's probably beyond the scope unless they ask. Just stick to the digits. Keep the answer simple and clear, avoiding any markdown as per the instructions.

Alright, final answer: first seven digits including the 3 are 3.141592, and the seven digits after the decimal are 1415926. But the user's question is about the first seven digits, so I'll present 3.141592 as the answer, then note the seven after the decimal if needed. Make sure to use the correct terminology so the user knows the difference.

Wait, let me check the sequence again. Pi is 3.1415926535... So the digits are 3, 1, 4, 1, 5, 9, 2. That's seven digits. If they want seven after the decimal, it's 1,4,1,5,9,2,6. But the initial seven digits are 3.141592. The user might be confused by the decimal point. Should I exclude the decimal point and just list the digits?

Yes, that's better. The first seven digits are 3141592. The seven digits after the decimal are 1415926. But the user asked for the first seven digits, so I'll go with 3141592. However, the question might be interpreted as the seven digits after the decimal. To be safe, I can present both but state the main answer clearly.

I think the best approach is to give the first seven digits as 3.141592, then note that the seven digits after the decimal are 1415926. That covers both bases. The user can choose based on their need.

So, the answer will be: The first seven digits of Pi are 3.141592. If you're asking for seven digits after the decimal, those are 1415926.

But let me confirm the count. Including the 3: positions 1-7 are 3,1,4,1,5,9,2. After the decimal: 1,4,1,5,9,2,6. Yes, that's correct.

```
I'll make sure the response is clear and addresses both possible
interpretations. The user might appreciate the thoroughness.

Also, maybe the user is not aware that Pi is approximately 3.14, but they
 want more digits. By providing both, I can help them understand the
precision needed.

In conclusion, the answer should be the first seven digits including the
3, which is 3.141592, and if they need seven after the decimal, it's
1415926.

I'll format the response accordingly, making it easy to read with the
digits listed clearly.

Alright, let me write the response now.
</think>
The first seven digits of Pi (including the integer part) are
**3.141592**.

If you're asking for the first seven digits *after* the decimal point,
they are **1415926**.
```

# I  PROMPTS

For Math500, AIME2025 and the free-form version of GPQA-Diamond, we used

```
Solve the following math problem efficiently and clearly. Please reason
step by step, and put your final answer within \boxed{}.
```

For the multiple-choice version of GPQA-Diamond, we used

```
Solve the following multiple choice problem efficiently and clearly.
Please reason step by step, and return one single option as the answer,
put your answer within \boxed{}.
```

When evaluating the open-ended version of GPQA-Diamond, we check the correctness usiong LLM as a judge following the prompt provided by Chandak et al. (2025)

```
Your task is to judge whether the given response to a question matches a
given ground truth answer or not. You are provided with a question, a
ground truth response, and the response you need to judge.
For a response to "match", it must have at least as much information as
the ground-truth.
The response can have more information than the ground-truth. It can be
more specific (for example, "Labrador" is more specific than "dog"), or
have additional possible correct answers. But it must cover everything
mentioned in the ground-truth. It is okay if it covers it in different
words, i.e. paraphrased.
For numeric answers, the relative error, defined as |response - ground
truth| / mean(response, ground truth), must be less than 1% for the
response to be judged as a correct match. Here, if the ground truth is a
specific numeric quantity but the response is a range, then they don't
match (even if the range contains the ground truth).

Possible judgments:

"0": The response does not match the ground-truth answer.
"1": The response matches the ground-truth.

Question: "{question}"
Ground truth: "{target}"
Response: "{response}"
```

```
Your job is to ONLY check whether the given response matches the ground
truth answer or not in the context of the question. You DO NOT NEED to
assess the correctness of the response. This is part of an automated
evaluation process, therefore you MUST OUTPUT your final answer as "0" or
 "1" in <answer> </answer> tags.

Think step by step and end your response with <answer>0</answer> OR <
answer>1</answer> TAGS.
```

