# OpenReview forum: "EAT: Entropy After $\textlangle \tt /Think \textrangle$ for reasoning model early exiting"
_ICLR.cc/2026/Conference — Submitted to ICLR 2026_

### Official Review · Reviewer_rXbq · 2025-10-30

**Soundness:** 3
**Presentation:** 3
**Contribution:** 3
**Rating:** 6
**Confidence:** 4

**Summary:**

This paper proposes a novel early exit method to address the overthinking problem in large-scale inference models. The authors first quantify this by tracking the Pass@1 metric, demonstrating that inference models often converge to the correct answer early in the inference chain, making subsequent token generation redundant. To address this, the paper proposes EAT (Entropy After </ think>) as a signal to monitor the inference process. This involves manually inserting </ think> tokens during inference and calculating the entropy of subsequent individual tokens to determine whether to exit early. Inference is terminated early when the variance of EAT, estimated via exponential moving average, falls below a threshold. Experiments show that this method can save 13-21% of token usage without sacrificing accuracy on datasets such as MATH-500 and AIME-2025, and it is also effective in black-box settings using a small proxy model to monitor a large inference model.

**Strengths:**

S1. The paper quantifies the "overthinking" phenomenon of the inference model through large-scale experiments (128 rollouts), which is clearly visible in Figure 1. The Pass@1 metric often saturates within the first 10-20% of the inference chain, yet the model continues to generate thousands of tokens. This finding is valuable because the output token cost of the inference model is far higher than the input token cost. In commercial APIs, output tokens are also typically more expensive. Therefore, the ability to adaptively allocate computational budgets has clear economic value and practical significance. The paper also promises to release over 20,000 GPU hours of inference trajectory data, a significant contribution to the community.

S2. EAT only needs to calculate the entropy of a single token after `<think>`. Compared to the baseline method, which requires generating a complete answer through rollout, EAT has a significant advantage of O(|R|) complexity, where |R| is the length of the reasoning chain, while the overhead of the rollout method is stochastic and increases linearly with the value of K.

S3.The paper finds that EAT can be calculated using small surrogate models (e.g., 1.5B parameters), thereby enabling early exit control of large inference models (e.g., 70B parameters). This is very important in practical applications.

S4. The paper's experiments covered multiple dimensions: three different inference models (DeepSeek-Qwen-8B, Llama-70B, Qwen-4B), four different datasets (MATH-500, AIME-2025, GPQA multiple-choice and open-ended), and two EAT computation methods (white-box and black-box). Figure 4 systematically compares EAT with the token-based baseline, showing the performance-efficiency trade-off curves at different thresholds, which is more convincing than single-point comparisons.

**Weaknesses:**

W1. Although the paper proposes a correlation between EAT and information gain in Equation 6, this relationship is heuristic rather than rigorously derived. The core question is: why can the entropy of a single token reflect the certainty of the entire answer? The paper does not provide a theoretical analysis. Observing Figure 2, we see a relatively smooth monotonically decreasing trend only at the manually labeled "Conclusion" position, suggesting that EAT may not be an essential, robust signal, but rather happens to be related to certain specific model behaviors.

W2. Appendix G.3 of the paper candidly presents two types of failure cases: unsolvable problems (Figure 11) and Pass@1 descent problems (Figure 12). For unsolvable problems, EAT is always unstable, and the algorithm will exhaust all T tokens; for Pass@1 descent problems, EAT may stop at a suboptimal position or also exhaust all tokens. However, the paper does not propose any potential solutions or improvement strategies for these failure cases. In real-world applications, we don't know in advance which problems are solvable and which will cause a decrease in Pass@1. If the method fails in these situations, its practical value is reduced. The paper should design a mechanism to detect these situations.

W3. The paper only includes a "solvable subset" in the GPQA results shown in Figure 4, meaning they pre-filtered out problems with a final Pass@1 > 0.8 for evaluation. This operation is methodologically problematic: in real-world applications, we cannot know in advance which problems are solvable; the method must be evaluated on the complete dataset. Pre-filtering is equivalent to running the method in "easy mode," which will naturally result in better performance.

W4. The paper criticizes the #UA@K method in Figure 5 for its high overhead due to the need to generate K rollouts. However, this criticism is unfair because the paper's own evaluation metric, Pass@1 (Avg@128), requires 128 rollouts. This presents a contradiction: the authors criticize the overhead of the baseline method, but their own ground truth metric has the same overhead.

W5. Using `\n` directly as a symbol in formula (5) is informal; a special symbol for newline should be used. The layout of the multiple subplots in Figure 1 can be optimized; the current column arrangement makes the legend and labels somewhat crowded. While the term "overthinking" is descriptive, it may not be formal enough in academic writing; expressions such as "redundant reasoning" or "reasoning inefficiency" could be considered. The warm-up condition `n ≥ 4/α` in Algorithm 1 lacks explanation and theoretical basis. The labels "Line 16," "Line 55," etc., in Figure 2 should be given more context in the text to help readers understand why these positions are special. Some experimental details (such as why a temperature of 0.6 and a top-p of 0.95 were chosen) could provide more motivation.

**Questions:**

Q1. Could you provide a more in-depth theoretical analysis to explain why EAT can serve as an effective signal for early stopping?

Q2. For the two types of failure cases shown in Appendix G.3 (unsolvable problems and Pass@1 descent problems), can an auxiliary mechanism be designed to detect and handle them?

Q3. Is it possible to compare the EAT and #UA@K methods under a fixed computational budget (e.g., the same total FLOPs, GPU time, or API call overhead)?

Q4. Could you report the performance on the full GPQA-Diamond dataset, and not just on a solvable subset?

Q5. The paper primarily uses DeepSeek and Qwen family models. Can the effectiveness of EAT be validated on other model families (such as Meta's Llama family, Anthropic models, etc.)?

---

> ### Author Response · Authors · 2025-11-21
> **Author response (1/)**
>
> We would like to thank the reviewer for the detailed and insightful feedback.
>
> Please see below for our responses to each point.
>
> ## (W1, Q1) Why does EAT work?
>
> Thanks for raising this very important point. We will mainly focus on answering this through your following great question
>
> > why can the entropy of a single token reflect the certainty of the entire answer?
>
> Assume the entire answer has N tokens, the N-gram entropy over an N-token autoregressive distribution, given a prompt $C$, is given by
>
> $$
> \mathbb{H}_{N}(C) = \mathbb{E}\_{T_1, \ldots, T_N \mid C} [-log P(T_1, \ldots, T_N \mid C)]
> $$
>
>
> where the expectation is taken with respect to all ${|V|}^N$ possible values ($|V|$ denotes the vocabulary size) in the conditional distribution
> $$
> P(T_1, ..., T_N \mid C) = P(T_1 \mid C) P(T_2 \mid T_1, C), ... ,P(T_N \mid T_1, ..., T_{N-1}, C)
> $$
>
> Importantly, we can decompose the N-gram entropy as
> $$
> \mathbb{H}_N(C) = \mathbb{H}\_{1}(C) +  \\
> \mathbb{E}\_{T_1 \mid C}[\mathbb{H}_1(C, T_1)] + \mathbb{E}\_{T_1, T_2 \mid C} [\mathbb{H}_1(C, T_1, T_2)] + \\
> \ldots  +  \mathbb{E}\_{T\_1, T\_2, \ldots, T\_{N-1} \mid C}  [\mathbb{H}\_1(C, T\_1, T\_2, \ldots, T\_{N-1})]
> $$
>
> where $\mathbb{H}_1(\texttt{context})$ denotes the unigram entropy over the next token after $\texttt{context}$.
>
> The decomposition is a summation of $N$ *positive* quantities, and **the first term of the decomposition, $\mathbb{H}\_{1}(C)$, corresponds to EAT**.
> This tells us that
> - *When the uncertainty over the longer horizon, $\mathbb{H}_N(C)$, starts to reduce and stabilize as reasoning proceeds, then EAT must also reduce and stabilize.*
> - *When EAT is at a high value and perturbs a lot, the longer horizon's uncertainty will also be high*
>
> However, note that our analysis does *NOT* answer the following three questions:
> - Whether the first token's stabilization *leads to* the stabilization of the whole rollout answer. Note that this may depend on the model's linguistic pattern, where the first token may serve as the indicator for "High-level plan", similar to the role of branching token proposed in recent literature [1]
> - Why does the uncertainty over the answer string reduce as reasoning proceeds? Some recent work observing this behavior, such as [2], and framing this as a "loss of diversity" issue, which can be problematic for e.g. creative writing. However, this behavior is also understandable in that post-training targets at maximizing Pass@1, which requires all rollout answer strings to contain one identical answer.
> - Why EAT correlates with Pass@1. Our observation indicates that when the model is confident, it is often correct, as shown by the high Pass@1. This seems to imply that the model is often well calibrated, i.e. the uncertainty level is informative about the correctness. On the other hand, when the model shows low Pass@1, it often has a high and noisy EAT trajectory. It is currently unclear to us which component in the post-training pipeline provides the model with such a feature.
>
> All three questions above are fundamental questions for understanding basic behavior of reasoning models beyond just EAT. While we think they are beyond the scope of the current manuscript, we will do better to explain the limitations of our understanding and connect it to these bigger questions and point to relevant literature where it exists.
> We do believe that our empirical observations, as well as the large-scale rollout we released, can motivate and accelerate future research along this line.
>
> [1] https://arxiv.org/abs/2506.01939
>
> [2] https://arxiv.org/abs/2509.02534

---

> > ### Author Response · Authors · 2025-11-21
> > **Author response (2/)**
> >
> > ## (W2, Q2) Solution for unsolvable problems
> >
> > This is again, a very good point. We believe the big question here is: What should we do when the EAT never stabilizes, i.e. the model fails to reach a confident answer, throughout the e.g., 10K token reasoning budget.
> >
> > We believe that understanding patterns that point toward non-stabilization and non-answerability is a useful direction for future work and that our work, including the exploration in the appendix, provides both motivation and inspiration for addressing this. We believe this will be separate from and complementary to EAT's early stopping when something appears answerable earlier.
> >
> > We believe that our work provides strong motivation and inspiration for exploring this direction. For example, per our results in Appendix G.3, Figure 11, 12 and 15, one may consider *variation in the EAT metric* as a signal of not making enough progress. We will point this out as valuable future work.
> >
> > Note that, unlike early stopping solvable problems, it is NOT sensible to use a fixed threshold when detecting unsolvable questions, since all questions could show high variance in the beginning; therefore, we should *set the criteria by measuring whether the model is making enough progress*.
> >
> > Lastly, it is worth noting that such regimes, despite the extra efficiency gain, will certainly increase the risk of stopping the reasoning **too early**, i.e. we stop the reasoning before the model reaches the correct answer, as it may still make a "break through" further down the thinking chain even if it may currently be stagnating. This is an inherent and unavoidable tradeoff.
> >
> > ## (W3, Q4) Full GPQA-Diamond results
> >
> > We have included the full results in Figure 20 in the appendix of the revised manuscript, where we notice that our method does not show *dataset-level token reduction*, since a majority of questions in the dataset are not solved by the model, where EAT never stabilizes.
> >
> > However, we believe this also reflects an important caveat of early stopping research: The performance depends on the distribution and composition of the dataset. If we consider an algorithm that targets cases where the model wastes tokens on solvable problems (e.g. our algorithm), but the dataset contains mostly unsolvable problems, then the algorithm will be less effective, though no worse than rolling out the full trajectory.
> >
> > ## (W4, Q3) Comparision with #UA@K
> >
> > > W4. The paper criticizes the #UA@K method in Figure 5 for its high overhead due to the need to generate K rollouts.
> >
> > We would like to clarify that, both #UA@K and EAT are *signals* used for checking when to stop, whereas Pass@1 (Avg@128) is the *label* that evaluates the model's correctness at the exit point for evaluation purposes.
> > The issue with #UA@K is that, in order to determine when to stop, you need to spend a lot of tokens to estimate the signal, which contradicts the motivation of early stopping. At deployment time, Pass@1 (Avg@128) will **NOT** be used anywhere because it requires access to the ground truth label.
> >
> > Notice that we choose to use 128 rollouts to estimate Pass@1 because the generation of the answer rollout is **random**, therefore, we choose to use a large number of rollouts to eliminate the randomness to ensure the reliability of our results.
> >
> > > Is it possible to compare the EAT and #UA@K methods under a fixed computational budget (e.g., the same total FLOPs, GPU time, or API call overhead)?
> >
> > In order to compare EAT with #UA@K under a fixed computational budget, we can reduce the frequency of evaluating #UA@K by at least $K \times T$ where $T$ denotes the length of the rollout.
> > We include experiments under this setting in Figure 19 in the appendix, where we evalute #UA@32 every 64 lines (note that the actual value of $T$ will be much larger than 2), and we find #UA@K shows suboptimal performance, worse than the token based baseline, because it measures the signal *too infrequently*.
> >
> > ## (Q5) Additional results on Claude
> >
> > > The paper primarily uses DeepSeek and Qwen family models. Can the effectiveness of EAT be validated on other model families (such as Meta's Llama family, Anthropic models, etc.)?
> >
> > Yes! In Figure 16 in the appendix, we show that EAT can also early stop reasoning from Claude 3.7 Sonnet from Anthropic, an API only model that returns users' full reasoning traces (rather than the summarized version), and we compute EAT locally using Qwen3-4B-Thinking-2507.
> >
> > These results also imply that the effectiveness of EAT is less sensitive to the behavior/structures in the reasoning chain, since EAT computed under the same model (Qwen4B) can be early to stop reasonings from different families of reasoning models, namely DeepSeek, Qwen and Claude models.
> >
> > Regarding Llama-based model, we are unaware of any official reasoning model from Meta. We are happy to conduct additional experiments if the reviewer can point us to any.

---

> > > ### Author Response · Authors · 2025-11-21
> > > **Author response (3/)**
> > >
> > > ## (W5)
> > > Thank you for noting these details! Please see below for our responses to each individual point
> > >
> > > - "A special symbol for newline should be used.": We use `\n` to help readers directly translate our equation to code they can implement themselves; however, we agree that this looks informal. We plan to replace `\n` with raw text `<EOL>`. Does this look like a better alternative to you?
> > >
> > > - Layout in Figure 1: We have increased the spacing between the columns in Figure 1 in the revised manuscript!
> > >
> > > - Overthinking (following prior art), we agree that the wording is informal. We choose this terminology to follow existing literature [1] rather than proposing new terms.
> > >
> > > - Warum up conditions: The warm-up condition $n \leq 4/\alpha$ in our Algorithm is a warm up period used to alleviate the initialization bias of the EMA, a trick widely used in practice, e.g. [3] suggests that "a warmup period of approximately 2.3 times the span size is required to reduce the initialization term’s weight to less than 1\% of the total influence.", where the span size (or effective window size) is approximately $2/\alpha$.
> > > However, in Figure 18 of the revised draft, we show that it is possible to use a debiased version of EMA to remove the warm-up phase.
> > >
> > > - The labels "Line 16," "Line 55," etc., in Figure 2: These correspond to the manually annotated lines where the models draw a conclusion.
> > >
> > > - Temperature and top-p: The parameters are the configurations recommended by the model releaser (https://huggingface.co/deepseek-ai/DeepSeek-R1-0528-Qwen3-8B#deepseek-r1-0528-1); we suspect that these are the parameters used during post-training time for generating rollouts.
> > >
> > > [1] https://openreview.net/forum?id=HvoG8SxggZ
> > >
> > > [2] https://arxiv.org/abs/2504.13367
> > >
> > > [3] https://terbe.dev/blog/posts/exponentially-weighted-moving-average
> > >
> > >
> > > ---------------
> > >
> > > We would like to thank you again for your efforts in providing the detailed and insightful comments. With the additional polish, clarification, and experiments we have incorporated, we find our manuscript to be a significantly better and clearer presentation of our contributions, thanks to your valuable feedback.

---

### Official Review · Reviewer_ftpa · 2025-10-31

**Soundness:** 3
**Presentation:** 3
**Contribution:** 3
**Rating:** 8
**Confidence:** 4

**Summary:**

This paper studies the next token distribution's entropy when the reasoning language model's thinking processes. The experiment results suggest a strong connection between the time when the model starts to overthink and the entropy goes down. Based on the observation, the authors introduce EAT to monitor the entropy signal, and stops thinking once it reaches a threshold. In experiemnts, EAT significantly reduces the token usage without accuracy drop, and can be extended to the black box models where the entropy cannot be accessed.

Compared to previous works on the reasoning model early exiting topic, EAT provides several advantages:
1. It does not require further training;
2. It does not introduce much redundant computation;
3. It supports black box models by computing EAT on a proxy model.

**Strengths:**

- The writing is easy to follow and well organized. The authors provide comprehensive experiments to naturally introduce the concept and effectiveness of EAT (and EMA of EAT), then perform experiments on using EMA of EAT for early exiting.
- The advantage of EAT (no need for further training, smaller redundant computation) is significant comparing to previous work.
- The authors provide evaluation of EAT on reducing reasoning tokens on several perspectives.

**Weaknesses:**

- The author provides several experiments to show the relation between EAT and the model's confidence. However, most of the examples in the paper are only for the positive cases where the task is easy: In Figure 1, the Pass@1 for the four sampled question all reaches 1. When the question itself is hard (Pass@1 after the whole thinking is less than 1 or even near 0), what is the relation between the EAT and Pass@1 / #Unique answers? Providing some data in this case may help improve the soundness of the motivation. The same question applies to Figure 2.
- The effectiveness of EAT against other early exiting methods (e.g. classifier based, sampling based) is not discussed in the experiments. Adding such baselines could help improve the significance of EAT.

**Questions:**

- When using a proxy model to estimate the EAT of a black box model, is it common that the computation of that proxy model can be even higher than the computation saved for the black box model?

---

> ### Author Response · Authors · 2025-11-21
> **Author response**
>
> We would like to thank the reviewer for the positive and encouraging feedback.
>
> Please see below for the responses to your concerns
>
> ## Negative examples for hard questions
>
> Thank you for raising this very important point!
>
> Figures 11, 12, and 15 (a new figure we included in the revised draft) in the appendix show some challenging questions from GPQA-Diamond and Math500, where the model consistently shows low Pass@1 across all 10K token budgets.
> Broadly, we observe that the model's uncertainty level, reflected by both EAT as well as #Unique Answers, stays at a high value and perturbs a lot, indicating that the model is "struggling" to reach a confident conclusion.
> In such cases, our approach would not perform any early stopping until we run out of budget, since EAT never stabilizes, and more reasoning may be needed for such examples.
>
> Notice that the beginning regions in each column in Fig.1 can also be viewed as a "local version" of hard questions; that is, the model cannot solve them in a small number of tokens.
> We can see that EAT and #Unique answers remain high and noisy early in the reasoning; however, later on, the model manages to converge to a confident answer as more reasoning is involved.
>
>
> ## Comparison with other approaches
>
> Token-based and #UA@K themselves are baseline approaches, where #UA@K can be viewed as a sampling-based approach; the comparison is shown in Figure 5.
>
> We additionally include another baseline proposed in [1] (which is currently under review at ICLR 2026 [2]):
> The approach first greedily samples an answer rollout, and then computes the averaged probability across the rollout tokens as the confidence score, which will be used as the signal for determining early stopping.
>
> The comparisons are shown in Figure 13 and Figure 14 in the appendix.
> Broadly, we find that EAT shows similar performance compared to the confidence proposed in [1] for early stopping, while being much cheaper, since EAT does not require rollouts, and has fewer hyperparameters, since EAT does not need to tune the decoding hyperparameter, such as the length and number of rollouts.
>
>
> [1] https://arxiv.org/abs/2504.15895
>
> [2] https://openreview.net/forum?id=NpU7ZXafRi
>
> ## Overhead of proxy model v.s. black box model
>
> > When using a proxy model to estimate the EAT of a black box model, is it common that the computation of that proxy model can be even higher than the computation saved for the black box model?
>
> This is again, a very good question! In short, when applied to black box models, we expect the computation of EAT to introduce **no extra time overhead** in practice.
>
> In particular, the models that need black box API style access are often **very large**, whose deployment requires sophisticated parallelism mechanisms across multiple GPU servers.
>
> In contrast, the proxy model is often **much smaller**, e.g., 1 - 8B parameters, which can be deployed on a single consumer-size GPU (< 32GB memory)
>
> And the idea is that the computation of EAT can **overlap** with the black box model's decoding process:
> ```
> Server side:     ...| Sampling chunk i  | Sampling chunk i + 1 | ...
> Local machine:   ...| EAT for chunk i-1 | EAT for chunk    i   | ...
> ```
> where each chunk is a number of tokens (e.g. 100 tokens) received from the API.
>
> Note that such overlap is feasible as long as the black box model is larger than the proxy model, in particular, given T reasoning tokens, it takes
>
> - O(BlackBoxModelSize * T) for sampling from the black box model, which is an unvectorizable, T-iteration operation.
> - O(ProxyModelSize * (T + 1)) for computing EAT, which involves prefilling the $T$ tokens' kv cache (vectorizable), and then computing the entropy.
>
> Therefore, computing EAT for a newly received chunk of text can be done while we are waiting for the next chunk of text.
>
> Our additional experiment results in the updated manuscript (Figure 16, Appendix) consider using a local Qwen4B to early stop Claude Sonnet 3.7 verify the feasibility, where we find that receiving a chunk of a 100-token text from the Claude API takes ~1 second, whereas computing EAT for the chunk takes less than 0.2 seconds on a single L40s GPU (Figure 17).

---

### Official Review · Reviewer_ijDx · 2025-10-31

**Soundness:** 3
**Presentation:** 3
**Contribution:** 2
**Rating:** 2
**Confidence:** 5

**Summary:**

The paper tackles overthinking in reasoning LLMs by proposing EAT (Entropy After \</think\>), a simple stopping signal that appends an end-of-reasoning marker (sometimes with an answer prefix) and measures next-token entropy. The key observation is that EAT stabilizes right around where Pass@1 accuracy plateaus, so they use an EMA + variance threshold to trigger early exit. On MATH-500, AIME-2025, and filtered GPQA, their offline simulations show about 6–22% token savings with minimal accuracy loss. They also show a smaller model can compute EAT as a proxy for a larger one. The signal is cheap and training-free, adapts to each instance, and the correlation with accuracy saturation is clear on R1/Qwen-style traces. However, the contribution is incremental because it remains within the same entropy/confidence-based halting paradigm, mainly shifting the probe to the token after an explicit stop and smoothing it, while omitting the most natural comparison point, single-pass answer-entropy halting. Also, the evaluation is post-hoc on single sampled traces, not live decoding. One other issue, which I trust to be a minor overlook from the authors is that some claims could be toned down. The abstract highlights 13–21% savings, but the actual results span 6–22% depending on task difficulty, only AIME hits about 21–22%.

**Strengths:**

- Clear Motivation and Problem specification
- Very simple, training-free method: one-token entropy after an explicit stop marker, plus EMA+variance sounds is easy to implement and cheap.
- Adaptive per-instance compute
- On AIME-2025 and similar structured tasks the method cuts about 13–22% tokens without hurting accuracy.

**Weaknesses:**

- The idea of using entropy for early stopping isn't new, but the paper doesn't compare against the most obvious alternatives, single-pass answer entropy or confidence-based stopping. These would be the most informative comparisons. This is an incremental contribution as it only adds EMA
- All the gains come from clean, math-heavy problems (MATH-500, AIME-2025) with models that already output \</think\> and structured answers. We don't know if this works on messier traces, shorter reasoning, or different model formats.
- Efficiency gains are computed offline from pre-generated traces, not measured during actual decoding. Real-world savings might differ.
- Claims like "first quantitative demonstration" seem strong given recent related work
- The method needs specific formatting (\</think\>, sometimes "Final answer:", certain newline patterns) and manual tuning (EMA warmup). Small changes in how the model structures output could break the signal.
- Savings do not generalise because on stronger models / harder datasets the reduction drops to about 6–11%, so the benefit isn’t consistent across settings.

**Questions:**

Refer to the weaknesses please and one more question:

Can you report how often your scheduled EAT checkpoints didn’t occur naturally (i.e. the model didn’t emit the expected \n\n / \</think\>), and what EAT did in those cases?

---

> ### Author Response · Authors · 2025-11-21
> **Author response 1/**
>
> We would like to thank the reviewer for the insightful feedback! Please see below for our response to each individual point.
>
> ## Novelty
>
> > The idea of using entropy for early stopping isn't new, but the paper doesn't compare against the most obvious alternatives, single-pass answer entropy or confidence-based stopping. These would be the most informative comparisons. This is an incremental contribution as it only adds EMA
>
> We agree that the idea of using entropy for early halting isn't new, as we discussed in the related work section: *Using uncertainty for adaptive computing is a widely adopted idea since the LSTM ages.*
>
> We also agree that there are very recent works (arxiv-ed ~May this year) that use uncertainty-inspired quantity for early stopping reasoning models; however, we are unaware of any existing approaches that claim to be "single-pass answer entropy or confidence-based stopping".
>
>
> Essentially, the answer for *free-form* questions would span across multiple tokens (e.g. $T$ tokens), and to compute either confidence (measured by maximum probability of the event on the support) or entropy, it would involve **enumeration over $|V|^T$ combinations**.
> Of course, it is possible to use Monte Carlo estimation to estimate the quantity, and if we use one sample to estimate this quantity, we can create a "single-pass answer entropy/confidence", is this what you are suggesting?
>
> The closest existing method we can find to this approach is the confidence score from [1] (currently under review at ICLR 2026).
> The authors of [1] suggest first generating rollouts after `</think> Final answer:`, then using the geometric mean of the maximum softmax probability [3] at each position of the rollout as the confidence score. In their appendix, the authors suggest generating one rollout using greedy decoding; therefore, the confidence score becomes the length-normalized likelihood over one rollout, which can be viewed as a *one-sample estimation of the multi-gram entropy* if we $\log$ it. (Note that this connection is not stated in [1])
>
> We therefore compare our method with [1], the results are shown in Figure 13 and Figure 14 in the appendix in the revised manuscript. Broadly, we find that EAT shows similar performance compared to the confidence proposed in [1] for early stopping (where we evalute EAT and confidence at same frequency), while being much cheaper, since EAT does not at all require rollouts, and has fewer hyperparameters, and since EAT does not need to tune the decoding hyperparameter, such as the length and number of rollouts.
>
> We are happy to incorporate more comparisons if the reviewer could point us to references that propose other single-answer confidence / entropy methods applied in the reasoning model domain.
>
> [1] https://arxiv.org/abs/2504.15895
>
> [2] https://openreview.net/forum?id=NpU7ZXafRi
>
> [3] https://arxiv.org/abs/1610.02136
>
>
> ## Structure of reasoning
>
> > All the gains come from clean, math-heavy problems (MATH-500, AIME-2025) with models that already output </think> and structured answers. We don't know if this works on messier traces, shorter reasoning, or different model formats.
>
> We have additionally included results evaluated on Claude Sonnet 3.7, whereas we use Qwen4B for computing EAT and early stopping remains effective.
>
> We believe the reasoning format of Claude is significantly different from the models we consider, i.e. DeepSeek and Qwen. Also, since Claude 3.7 Sonnet is released Feb 2025, later than DeepSeek R1, it is unlikely that DeepSeek is distilled from Claude.
>
>
> ## Measurement of efficiency gain
> > Efficiency gains are computed offline from pre-generated traces, not measured during actual decoding. Real-world savings might differ.
>
> Note that we believe the savings in tokens measured post hoc can directly translate to savings in wall clock time, since decoding *is a sequential process*, i.e. tokens need to be generated one by one. Additionally, since evaluating EAT only incurs *minimal overhead* (close to generating one token), if we incorporate the overhead of EAT into account by viewing each EAT evaluation as using one extra token, *we still use much fewer tokens to achieve the same performance compared with token based baseline* (Figure 21 in revised manuscript)
>
> Additionally, in the black box API model setting, the computation of EAT can overlap with the generation of the reasoning text, so EAT will not introduce any additional walk clock time overhead. In our additional experiments with Claude Sonnet 3.7, where we demonstrate significant wall clock time reduction by using EAT (at least 60 seconds per query)

---

> ### Author Response · Authors · 2025-11-21
> **Author response 2/**
>
> ## Robustness of our approach
>
> > The method needs specific formatting (</think>, sometimes "Final answer:", certain newline patterns) and manual tuning (EMA warmup). Small changes in how the model structures output could break the signal.
>
> Notice that the special formatting requirement is *mainly required for the model that computes EAT*. Our empirical results show that the same format for computing EAT is effective on reasoning traces from different models and families.
>
>
> The usage of EMA warmup is to alleviate the initialization bias for EMA: Since we initialize the EMA estimation of variance as 0, which could cause underestimation of the variance in the beginning of the reasoning. However, in Sec G.8 and Figure 18 in the revised manuscripts, we show that the warmup phase can be removed by using a debiased version of EMA (similar to the one used in Adam), and the performance gain remains.
>
> ## Stronger model
>
> > Savings do not generalise because on stronger models / harder datasets the reduction drops to about 6–11\%, so the benefit isn’t consistent across settings.
>
> We would like to clarify that DeepSeek-R1-Distill-Llama-70B is a larger model but NOT a **stronger** model, reflected by its accuracy of 0.82 v.s. 0.85 from DeepSeek-R1-0528-Qwen3-8B on Math500.
> The reason why the reduction is smaller is that the model generates shorter thinking: Under a maximum sampling budget of 10K tokens per question, DeepSeek-R1-Distill-Llama-70B uses only ~$9 \times 10^5$ for all 500 questions, while DeepSeek-R1-0528-Qwen3-8B uses ~$3 \times 10^6$ tokens in total.
>
> ## Frequency of computing EAT
>
> > Can you report how often your scheduled EAT checkpoints didn’t occur naturally (i.e. the model didn’t emit the expected \n\n / </think>), and what EAT did in those cases?
>
> The `</think>` token is not something emitted by the model; it is something we manually append to compute EAT. Notice that the majority of the recent reasoning models have some special tokens or formats that denote end of thinking, most models use `</think>`, except for GPT-OSS, which has a "analysis channel" field in its Harmony chat template (https://cookbook.openai.com/articles/openai-harmony#channels).
>
> Regarding the `\n\n`, we discussed this in line 305 - 310 in the original manuscript (right below Figure 3) in the original manuscript, note that this is not a requirement, we chose to compute EAT at new line because we compute the expensive ground true accuracy metric, pass@1, at `\n\n`, therefore we also compute EAT at these positions for implementation easiness. In appendix E, we evaluated EAT every tokens, and we EAT remains highly correlated to the Pass@1, and we do not see any reason the early exiting performance would drop in such cases.
>
> Lastly, in our experiments of early stopping the Claude model, we did NOT utilize any string pattern or special characters to decide when to compute EAT; we simply compute EAT every time we received 100 text blocks from the API without looking at the contents in the text, and early stopping remains effective.

---

> > ### Author Response · Authors · 2025-11-26
> >
> > Dear reviewer ijDx
> >
> > We'd like to thank you again for your helpful comments. The public discussion phase will end in a week, so please do let us know if our response addresses your concerns and questions.
> >
> > In particular, your review indeed helps us to build a deeper understanding regarding the performance-wise differences between EAT vs. existing approaches for computing the confidence!
> >
> > It would be very nice if you could let us know if our arguments look sensible and convincing to you! Any additional feedback is appreciated!
> >
> > Best, Authors of EAT

---

### Official Review · Reviewer_tdPT · 2025-10-31

**Soundness:** 3
**Presentation:** 3
**Contribution:** 2
**Rating:** 4
**Confidence:** 4

**Summary:**

The paper propose a novel early stopping mechanism by measuring the entropy after </think> (EAT) as a reliable confidence measure to guide the model early stopping. Specifically, EAT first measure the `[prefix]<think></think>` entropy, and monitor at every reasoning step  the entropy of `[prefix]<think>r1 r2 ... rn</think>` to obtain a its variance under an exponential moving average. If such value passes a threshold, the reasoning process can practically early stop with a good result. The evaluation and ablation shows good result of the method across different models and datasets.

**Strengths:**

1. The methodology (EAT) is sound.
2. The paper evaluate EAT on open-ended questions and show appealing result with early stopping.
3. The writing is easy to follow.

**Weaknesses:**

1. The solution lacks novelty. The paper needs to make a stronger and thorough comparison between EAT and the other methods to demonstrate its novelty.
2. EAT may prematurely terminate reasoning when the model has not yet arrived at a correct answer. This could limit the model’s opportunity for self-correction and increase the risk of incorrect outputs.

**Questions:**

1. The paper claims to be "first quantitative demonstration that reasoning models frequently overthink, continuing to generate lengthy reasoning chains even after the prediction has stabilized", which is not quite true given work mentioned in related work such as [Fu et al.](https://openreview.net/forum?id=wpK4IMJfdX, https://arxiv.org/abs/2412.20993), Yong et al. (https://arxiv.org/pdf/2505.18237), and other quantitative works using entropy and similar methods to control the reasoning trajectory. I would suggest to strengthening the statement with tailored comparison against these works, or soften the statement.
2. The paper choose to use exponential averaging and hyperparameter window size $\alpha$ and a predefiend threshold $\delta$.
3. Does the method prematurely terminate reasoning when the model has not yet arrived at a correct answer? This could limit the model’s opportunity for self-correction and increase the risk of incorrect outputs. The paper would benefit from a more detailed discussion of such failure cases and how EAT behaves under different reasoning dynamics.
4. EAT does not stabilize when the question is unsolvable, meaning EAT will waste all the tokens, which seems to be a big limitation of the method. Is there any mitigation the authors want to propose?

---

> ### Author Response · Authors · 2025-11-21
> **Author response 1/**
>
> ## Claim of our paper
>
> We would like to clarify that we are NOT claiming to be the first paper that points out overthinking, but rather one *new way to describe overthinking behavior* and a corresponding algorithm to stop it early. We will certainly qualify this statement and contextualize our work better in the literature, as detailed below.
>
> Our statement on "first quantitative demonstration that reasoning models frequently overthink" is centered around the top two rows in Figure 1, which aims at understanding the **dynamics** of the elicited predictive distribution throughout reasoning (i.e. p(answer | question, reasoning)). Note that since the answer parts for free-form questions are usually of unknown length with unknown closed form distribution, the easiest way to understand their statistical properties is to independently **generate a large number of rollouts from this distribution** and study its empirical properties.
>
> We believe our observation (top two rows of Figure 1) tells us a more detailed picture of overthinking, in addition to just "The model can *solve* the question without using all reasoning tokens", in particular:
> - In the beginning, the model is not capable of coming up with a correct answer. (Pass@1 ~ 0)
> - After some reasoning, the model can answer correctly *by chance* (0.1 < Pass@1 < 1.0)
> - As reasoning proceeds, the model starts to always answer the question correctly (place for early stop), i.e. the model becomes fully confident in one single answer.
>
> The observation indicates that reasoning gradually increases the model's chance of generating only a single answer, which is more often correct in one shot, i.e., *eliminating* the uncertainty in the predictive distribution, confirming the motivation for using confidence/entropy for early stopping.
>
> Other works analyze overthinking in complementary ways:
> - Figure 1 of [3] and Figure 2 in [1, 2] look at a single rollout's correctness throughout reasoning.
> - Figure 3 in Yong et al. ([4]) shows that the model's uncertainty reduces throughout reasoning, but only in the multiple choice QA setting rather than free form generation problems.
>
> However, we agree that this statement may cause confusion, and we have revised the statement as:
>
> **"We are the first work that quantitatively shows overthinking from a distribution dynamics perspective"**
>
> We hope this qualification is more appropriate, and we are certainly open to further dialogue to refine this and make it correct and appropriate. We thank you for pushing us to qualify our statement more carefully and contextualize our work.
>
> [1] https://arxiv.org/abs/2412.20993
>
> [2] https://openreview.net/forum?id=wpK4IMJfdX
>
> [3] https://arxiv.org/abs/2504.15895
>
> [4] https://arxiv.org/pdf/2505.18237

---

> > ### Author Response · Authors · 2025-11-21
> > **Author response 2/**
> >
> > ## Novelty and comparison with existing methods
> >
> > There are two main differences between EAT and existing methods:
> > 1. EAT does not require generating rollouts for estimating uncertainty: Note that all existing works require sampling rollouts for estimating the uncertainty, which introduces additional hyperparameters, such as the number of samples, the decoding hyperparameters, and it is unclear to us what the best practices are to choose these hyperparameters (we did not find related discussion or ablation study in prior works). In contrast, EAT is, by design, a deterministic quantity given the reasoning chain that does not have such parameters and (in its vanilla form proposed in the main text) only looks at the distribution of only one more token and so is quite cheap computationally.
> > 2. EAT can be applied to black box models without logit access: Our approach works with black box API models that do not expose log probabilities, where EAT can be computed using a proxy model, and in our additional experiment results, we further demonstrate using EAT to early exit Sonnet 3.7 from Claude (Figure 16, appendix). We are not aware of any prior works studying this capability.
> >
> > Perhaps coincidentally, the applicability of early stopping approach for black box models is raised by one of the anonymous reviewers for [2] (See weakness 1 from https://openreview.net/forum?id=NpU7ZXafRi&noteId=viBnQQtrx3), which reflects the community's need for such capability.
> >
> > (None of this submission's authors participated in any way in the review process of [2])
> >
> >
> > ### **Performance comparison with confidence score**
> >
> > We agree that direct comparisons with existing methods can help practitioners better understand the picture. Therefore, we additionally include the confidence signal proposed in [1] as a baseline (the work is currently under review at ICLR 2026 [2]):
> > The approach first greedily samples an answer rollout, and then computes the averaged probability across the rollout tokens as the confidence score, which will be used as the signal for determining early stopping.
> >
> > The comparisons are shown in Figure 13 and Figure 14 in the appendix of the revised manuscript.
> >
> > Broadly, we find that EAT shows *similar* performance compared with the confidence proposed in [1] for early stopping, while being *much cheaper*, since EAT does not require rollouts.
> >
> > [1] https://arxiv.org/abs/2504.15895
> >
> > [2] https://openreview.net/forum?id=NpU7ZXafRi

---

> > > ### Author Response · Authors · 2025-11-21
> > > **Response 3/**
> > >
> > > ## Window size and threshold (Question 2)
> > >
> > > > The paper choose to use exponential averaging and hyperparameter window size \(\alpha\) and a predefiend threshold \(\delta\).
> > >
> > > We indeed use window-size and threshold parameters. We could not see a question in the reviewer's comment, however. We would appreciate it if the reviewer could further clarify the question.
> > >
> > > ## Premature termination and unsolvable question
> > >
> > > Thanks for raising these two very insightful questions! We believe these two questions are actually highly correlated, so we will address them together.
> > >
> > > > Does the method prematurely terminate reasoning when the model has not yet arrived at a correct answer?
> > >
> > > In theory, there is a chance that EAT will prematurely terminate for some examples. In fact, there is always a performance and cost trade-off when controlling the reasoning token budget, which is the main feature that enables the "test time scaling" curve. Imagine the model can arrive at the correct answer at 20K reasoning tokens but we limit the budget to be 10K tokens, then we are essentially terminating the reasoning too early.
> > >
> > > However, EAT achieves a better trade-off compared to other methods thanks to our adaptivity, and our results in Figure 4 illustrate this: EAT uses fewer tokens compared with token-based early stopping (i.e., fixing the number of reasoning budget for each question) while maintaining the same accuracy, suggesting that EAT has a better trade-off and "spends" the token budget wisely where it is actually needed.
> > >
> > >
> > > > EAT does not stabilize when the question is unsolvable, meaning EAT will waste all the tokens, which seems to be a big limitation of the method. Is there any mitigation the authors want to propose?
> > >
> > > Note that another way to look at this behavior is:
> > > **EAT does NOT take the risk of terminating this prematurely since it is possible that the model can arrive at a correct solution with more reasoning**, so although it seems to waste all the tokens in the given budget, potentially more reasoning may well solve the question. This is an inherent tradeoff. We still do not think longer than the original model would, however, as we do not suppress the natural generation of the end-think token.
> > >
> > > Nonetheless, we do believe that understanding patterns that point toward non-stabilization and non-answerability is a useful direction for future work. We believe this will be separate from and complementary to EAT's early stopping when something appears answerable earlier. More than that, we believe that our work provides strong motivation and inspiration for exploring this direction. For example, our experiments in Appendix G.3 (as well as Figure 15) very transparently study the limitations of our approach in these settings, one may consider *variation in the EAT metric* as a signal of not making enough progress. We will point this out as valuable future work.
> > >
> > > Note that, unlike early stopping solvable problems, it is NOT sensible to use a fixed threshold when detecting unsolvable questions, since all questions could show high variance in the beginning; therefore, we should set the criteria by *measuring whether the model is making enough progress*.
> > >
> > > --------------------
> > >
> > > We do hope our responses address your concerns! We are open to continuing dialogue to explore the important points you raise and provide further clarifications. Thank you again for your efforts.

---

> ### Comment · Reviewer_tdPT · 2025-11-21
>
> Thank you for qualifying the statement.
>
> *Claim of our paper*
>
> > Figure 1 of [3] and Figure 2 in [1, 2] look at a single rollout's correctness throughout reasoning.
>
> From my reading, Figure 2 in [1, 2] appears to track correctness over 10 rollouts as the model consumes more reasoning tokens. In the heatmap, each cell corresponds to: “after spending X tokens, for the Y-th question, how many of the sampled rollouts have produced the correct answer so far?” In that sense, the visualization seems conceptually similar to what the authors here aim to study.
> I do however agree Figure 1 of [3] only examines a single rollout at a time. And likewise, Figure 3 in Yong et al. ([4]) does not extend to free-form generation settings.
>
>
> Where I may be misunderstanding is the distinction the authors are drawing between analyzing “distribution dynamics” versus the perspective offered in Figure 2 of [1, 2]. I would welcome a bit more clarification on what specifically differentiates these viewpoints. Strengthening this part of the writing might help readers more clearly understand the novelty in how the authors characterize overthinking.
>
> I am open to being convinced and would be glad to help refine the framing if additional explanation is provided.

---

> ### Author Response · Authors · 2025-11-21
>
> We would like to thank the reviewer for the prompt response
>
> > From my reading, Figure 2 in [1, 2] appears to track correctness over 10 rollouts as the model consumes more reasoning tokens.
>
> Per our understanding, the rollouts for Figure 2 in [1, 2] refer to full *reasoning trajectories*, instead of answer rollouts at a step of reasoning, as evidenced by the caption of Figure 7 in [1], the authors seem to have collected 10 reasoning chain for each question.
>
> We didn't find any explicit discussion on this in the paper, but if we look at their code, they only seem to generate **one answer** at each token limit, and the loop over "trial" is for generating an independent reasoning chain:
> See https://github.com/hao-ai-lab/Dynasor/blob/main/benchmark/TokenDeprivation/post_process.ipynb
> ```
> def earlyexit_accuracy(problem_sets, problems, jump = 4, continue_certain_bar = 2, warmup_steps = 0, max_tokens=limit, response_len=10, skip_wait=False):
>     all_scores = []
>     all_token_counts = []
>     tokens_per_problem_clip = [[] for _ in range(problems)]
>     corrects_per_problem_clip = [[] for _ in range(problems)]
>     score_dict = defaultdict(lambda: 0)
>
>     for run in range(trials):
>         token_counts = []
>         scores = []
>         for problem_id in range(problems):
>             ended = False
>             end_step = 100000
>             for step in range(0, max_tokens, step_size):
>                 step_id = step // step_size
>                 finished = problem_sets[f"{problem_id}_{run}"]['is_finished'][step_id]
>                 score = problem_sets[f"{problem_id}_{run}"]['score'][step_id]
>          ...
> ```
> as well as the usage of "num_trials" in their main experiment file
>
> https://github.com/hao-ai-lab/Dynasor/blob/main/benchmark/TokenDeprivation/run.py#L100
>
> https://github.com/hao-ai-lab/Dynasor/blob/main/benchmark/TokenDeprivation/run.py#L106
>
> So in summary, Figure 2 in [1, 2] verifies if the observation that "model can solve the question without using all tokens" is consistent across independently sampled reasoning chains.
>
> This is a different perspective than our illustration, which shows how the elicited predictive distribution evolves *within a single reasoning chain*, captured by generating 128 answer strings at a single step.
>
> -------------
>
> It is also worth noting that we have trouble finding the detailed definition of "accuracy" in many related work; that's why we define our metric (Eq.8) in a very explicit and perhaps verbose way.
>
> [1] https://arxiv.org/abs/2412.20993

---

> > ### Comment · Reviewer_tdPT · 2025-11-25
> >
> > Thank you for the comprehensive response. I am satisfied that the authors have addressed all of my comments, and I have updated my score to acknowledge the revisions.

---

> > > ### Author Response · Authors · 2025-11-25
> > > **Thank you for the response!**
> > >
> > > Dear reviewer tdPT
> > >
> > > *We have revised the manuscript, line 82 - line 83, to refine the contribution of our work* per our discussion.
> > >
> > > We would like to thank you again for your helpful insights and your updated scores for acknowledging the value of our work.
> > >
> > > Best,
> > >
> > > Author of EAT

---

### Author Response · Authors · 2025-12-01

# General Response

We would like to thank all the reviewers as well as the AC, SAC, and the program committee for their time, effort, and commitment throughout the review and discussion period.
We also feel deeply sorry that we can no longer continue the conversation with the reviewers due to the disruption, as the discussions so far have been highly constructive and invaluable for improving our work.

To summarize, we propose **EAT (Entropy After `</Think>`)**, a simple, training-free, and computationally inexpensive signal for adaptive early exiting in reasoning models.
By appending a stop-thinking marker (`</think>`) and monitoring the entropy of the next single token, we obtain a trajectory whose stabilization strongly correlates with when Pass@1 saturates, allowing compute to be allocated adaptively on a per-instance basis.

Compared with existing early exiting approaches, EAT is
- **Rollout free**: EAT does not require sampling rollouts for computing the uncertainty signal, making EAT cheap, deterministic, and having significantly fewer hyperparameters.
- **Applicable to black box models**: When the reasoning models' logits are not accessible (e.g. large, API only models), we can compute EAT using a local small surrogate model. During the rebuttal phase, we further demonstrate early stopping API only Claude Sonnet 3.7 using a local Qwen4B.

The novelty and value of our contributions were recognized by the reviewers as:

"...Very simple, training-free method...easy to implement and cheap..." (`ijDx`)

"The advantage of EAT (no need for further training, smaller redundant computation) is significant comparing to previous work." (`ftpa`)

"...the ability to adaptively allocate computational budgets has clear economic value and practical significance..." (`rXbq`)

"EAT can be calculated using small surrogate models...enabling early exit control of large inference models...This is very important in practical applications." (`rXbq`)

## Summary of discussion

Due to the incident, we are only able to complete discussions with reviewer `tdPT`. Reviewer `tdPT` initially raised concerns about the phrasing of our contribution claim, lacking comparison with existing works, whether our approach would stop reasoning prematurely, and mitigation for unsolvable questions.

We adjusted and refined the contribution statement, included additional baselines, and proposed a potential solution for detecting unsolvable questions. Reviewer `tdPT` found our responses satisfied and updated the score from 4 to 6 accordingly (https://openreview.net/forum?id=hfEVqiJyF6&noteId=Y3ttueAIgW).

Reviewer `ijDx` suggested comparing EAT with "single-pass answer entropy". While they did not provide any further specifics and did not respond, we made sure to compare to baselines along those lines and explained what we did.

## Summary of General Concerns and Corresponding Changes

Below, we provide a summary of the common concerns raised by multiple reviewers, along with the changes we have made in response:

- **Comparisons with baseline**: Reviewer `tdPT`, `ijDx` and `ftpa` asked for comparisons with existing baseline approaches. We have included [comparisons](https://openreview.net/forum?id=hfEVqiJyF6&noteId=AZPISNc4lm) with the confidence score proposed in [1] (concurrently under review at ICLR 2026 [2]).
- **Unsolvable question**: Reviewer `tdPT`, `ijDx` and `rXbq` asked for mitigation for unsolvable questions. We provided a [solution](https://openreview.net/forum?id=hfEVqiJyF6&noteId=CRJC8dlTES) based on the variation in EAT trajectory.
- **Warmup period of EMA**: Reviewer `ijDx` and `rXbq` suggested that the warmup length of the exponential moving average (EMA) may require manual tuning. To address the concern, we introduced a warmup free version of our early exiting algorithm in the revised manuscript (Figure 18) based on a debiased version of EMA.


[1] https://arxiv.org/abs/2504.15895

[2] https://openreview.net/forum?id=NpU7ZXafRi

---

### Meta-Review · Area_Chair_kSe9 · 2026-01-23

**Summary:**

This paper proposes a training-free early stopping method EAT (Entropy After </think>) to tackle the overthinking problem in reasoning traces, where correct answers can be generated much earlier. EAT inserts end-of-reasoning markers </think> (sometimes with an answer prefix) during inference and tracks the entropy of their subsequent tokens by maintaining an exponential moving average of their variance. The key observation is that EAT stabilizes right around where Pass@1 accuracy plateaus, so they stop inference early when the variance falls below a threshold. On MATH-500, AIME-2025, and filtered GPQA, their offline simulations show about 6–22% token savings for different models with minimal accuracy loss. They also show that EAT computed on a smaller open-source model can determine early stopping on a larger model.

**Reviewer Concerns:**

- The novelty of using token entropy to determine when to stop is limited due to several previous/concurrent works with similar ideas.
- Since the main analysis is conducted on solvable, well-structured, clean math reasoning traces, it is not clear whether EAT still performs stably on challenging unsolvable tasks, messier traces, shorter reasoning, or different model formats.
- There are a few overclaims in the paper, given some potential baselines and related works that were not discussed in the paper.
- The gain on different models varies and becomes modest on larger models, which weakens the method's importance.
- The robustness of EAT to several hyperparameters and warm-up settings for EMA was not justified.
- The claim of advantages in efficiency is questionable, as the proposed method is also evaluated based on multiple rollouts.
- Lack of comparison with baselines, such as lightweight classifiers on hidden states and confidence.
- In-depth or theoretical analysis cannot be found to understand why the single token entropy can indicate the correctness of the whole answer.

**Reviewer Scores:**

- The initial review ratings are 2, 4, 6, 8, with confidence of 5, 4, 4, 4, making it a borderline paper.
- After the rebuttal, only the reviewer who gave a rating of 4 participated in the discussion and raised the rating from 4 to 6.
- Given the lack of review feedback, I carefully read the paper and all comments.
- While the rebuttal did address some major concerns, I found that a few key concerns have been only partially addressed, for example, classifier-based baselines are still missing, the limitation of EAT when applied to unsolvable problems or irregular reasoning traces "in the wild", and the efficiency advantage given that multiple rollouts are required.
- I agree with several reviewers that the main idea of using token entropy to determine when to stop and generate a final answer is not new, and the paper overclaims some contributions. This may hurt the novelty.
- Therefore, I lean towards not accepting the paper. However, three out of four reviewers finally voted for acceptance. So I will leave the final decision to SPCs and PCs.

---

### Decision · Program_Chairs · 2026-01-26

Reject